# Collagen VI microfibril structure reveals mechanism for molecular assembly and clustering of inherited pathogenic mutations

Alan R. F. Godwin [1], Mark H. Becker[1,3], Rana Dajani[1,3], Matthew Snee [1,3], Alan M. Roseman [2] & Clair Baldock [1] ✉

Collagen VI links the cell surface to the extracellular matrix to provide mechanical strength to most mammalian tissues, and is linked to human diseases including muscular dystrophy, fibrosis, cardiovascular disease and osteoarthritis. Collagen VI assembles from heterotrimers of three different α-chains into microfibrils, but there are many gaps in our knowledge of the molecular assembly process. Here, we determine the structures of both heterotrimeric mini-collagen VI constructs and collagen VI microfibrils, from mammalian tissue, using cryogenic-electron microscopy. These structures reveal a cysteine-rich coiled coil region involved in trimerisation as well as microfibril assembly. Furthermore, our structures show that pathogenic mutations are located at interaction sites involved in different steps of collagen VI assembly, from the trimeric-coiled coil region that mediates heterotrimerisation, to clusters of mutations in the triple-helical region involved in microfibril formation. Our microfibril structure provides a template for understanding supramolecular assembly, and offers a platform for rationale design of therapeutics for collagen VI pathologies.

Collagen VI is a ubiquitously expressed collagen that forms heteromeric microfibrils with a beads-on-a-string appearance[1]. Collagen VI is found in most mammalian tissues, including cartilage, skeletal muscle, cardiovascular, kidney, cornea and skin. Although there are six collagen VI α-chains[2–4], in humans, the α4 chain is not expressed and the most common heterotrimer composition is the α1,α2,α3 heterotrimer. Mutations in collagen VI α1, α2 and α3 chains mainly affect the musculoskeletal system, leading to diseases such as Bethlem myopathy, Ullrich congenital muscular dystrophy (UCMD) and osteoarthritis (OA)[5–7]. Collagen VI is a key component of the skeletal muscle stem cell niche and, through regulating matrix stiffness, is involved in preserving stemness features of adult muscle stem cells[8].

A GWAS study has identified a COL6A3 variant as one risk locus for thoracic aortic aneurysms and dissections[9] and two variants in COL6A1

have been identified in some individuals with Spontaneous Coronary Artery Dissection, although their pathogenic significance is not clear[10]. Increased expression of collagen VI genes is frequently observed in cancer[11], in particular the α3 chain, where protease cleavage releases the matrikine endotrophin[12,13]. Collagen VI acts as an anchor between cell surface receptors, such as integrins[14,15] and the matrix and has a role in mechanotransduction in tendon, which is disrupted in UCMD[16]. Collagen VI has been shown to interact with many matrix components, including decorin, collagen II, collagen IV, aggrecan and fibronectin[17,18]. This link between the cell surface and the matrix has been shown to be cytoprotective, as disruption of collagen VI causes an increase in apoptosis through perturbation of autophagy[19,20].

All collagen VI chains have a short collagenous region flanked by globular N and C-termini, which are mainly composed of domains

[1]Division of Cell-Matrix Biology and Regenerative Medicine, Manchester Cell-Matrix Centre, School of Biological Sciences, Faculty of Biology, Medicine and Health, Manchester Academic Health Science Centre, University of Manchester, Manchester, UK. [2]Division of Molecular and Cellular Function, School of Biological Sciences, Faculty of Biology, Medicine and Health, Manchester Academic Health Science Centre, University of Manchester, Manchester, UK. [3]These authors contributed equally: Mark H. Becker, Rana Dajani, Matthew Snee. ✉e-mail: clair.baldock@manchester.ac.uk

homologous to the A-domains of Von Willebrand factor (vWA)[2] (Fig. 1a). Chains α1 and α2 are similar in size and domain structure, whereas the α3 chain is much longer[2]. The α4, α5 and α6 chains have similarities to the α3 chain; each of these chains contains seven N-terminal vWA domains[3]. Collagen VI assembles in a hierarchical manner where three collagen VI α-chains associate to form heterotrimers[1]. This requires the combination of the short α1 and α2 chains[21] with one long α-chain (α3, α5 or α6 chain) in an α1α2αX combination, where X is any long α-chain, which can be interchangeable[3,4]. The sequence and structural requirements for heterotrimerisation are currently undefined. These heterotrimeric assemblies (termed collagen VI monomers) are stabilised by the formation of interchain disulphide bonds[1] (Fig. 1b).

Collagen VI monomers interact in a staggered anti-parallel arrangement to form disulfide-bonded dimers, then two parallel dimers form tetramers, which are thought to be the form that is secreted into the extracellular space. The end-to-end assembly of tetramers[1] forms beaded microfibrils, which have globular regions separated by triple-helical collagenous regions[22]. The C-terminal domains of each chain are thought to be involved in higher-order assembly[23] and at least five N-terminal vWA domains are required for microfibril assembly[24], but the molecular details of collagen VI assembly are unknown. Collagen VI dimers and tetramers are homotypic, containing only one type of long chain; however, heteromeric microfibrils may form from different long chains[25].

Here we report the cryo-electron microscopy (cryoEM) structures of a heterotrimeric collagen VI C-terminal fragment recombinantly expressed in mammalian cells and the bead region of microfibrils extracted from mammalian tissue, to 3.14 Å and 4.33 Å resolution,

respectively. These structures allow us to model the individual α-chains within the microfibril bead region and to identify a cysteine-rich coiled coil region, which appears important for heterotrimerisation. To test the function of this coiled coil region, we mutate key hydrophobic residues in the mini-heterotrimeric collagen chains, which prevent the formation of a disulphide-bonded heterotrimer. Our microfibril structure allows us to map the locations of UCMD disease-causing mutations and we show a pathogenic hotspot in the tetramer interface important for microfibril assembly.

## Results

### An expression system for heterotrimeric collagen VI production

Collagens assemble from a C-to-N-terminal direction, often due to a trimeric C-terminal non-collagenous domain that initiates triple-helical assembly[26]. However, collagen VI does not have the trimeric C-terminal ColFi (NC1) domain found in fibrillar collagens and is a heterotrimer of three different α-chains. Therefore, to understand how collagen VI monomers assemble, an expression system was designed to generate mini-collagen VI heterotrimers with the C-terminal domains of the α1, α2 and α3 chains and a short stretch of collagenous sequence with nine Gly-X-Y triplets (Fig. 2a). Various α3 chain constructs were made including the full C-terminal region α1α2α3^FL, and deletion constructs only containing the C1 (α1α2α3^C1) or C1 and C2 vWA domains (α1α2α3^C1C2)(Supplementary Fig. 1). For the α1α2α3^C1C2 and α1α2α3^FL constructs, the furin cleavage site between the C1 and C2 domains in the α3 chain was mutated to prevent cleavage (RDRR to RDAA). N-terminal 6x His, Twin-Strep or FLAG-tags were introduced into the α1, α2 and α3 chains, respectively, and the constructs were co-transfected into HEK Expi293F cells. Heterotrimers were purified using

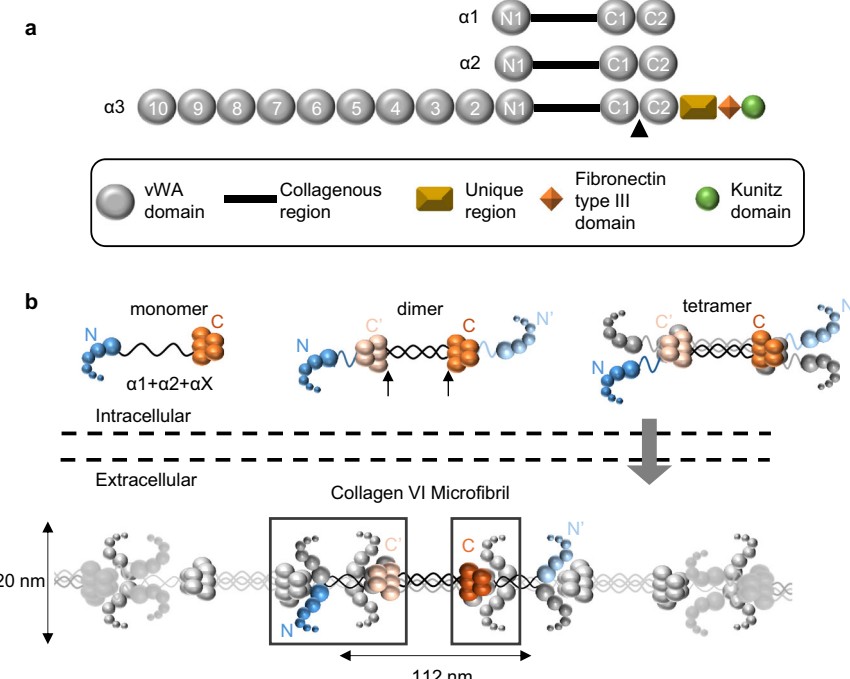

**Fig. 1 | Domain organisation and microfibril formation of collagen VI. a** Domain arrangement of the three most common collagen VI α-chains. The C- and N-terminal vWA domains are numbered from the collagenous region outwards. The black arrowhead indicates a furin cleavage site between the C1 and C2 domains in the α3 chain. **b** Cartoon representation of collagen VI microfibril assembly. Collagen VI heterotrimeric monomers form from one α1, one α2 and one αX chain, where X can be any of the long α-chains. The C-terminal globular regions are shown in orange and the N-terminal domains are shown in blue. For clarity, the C-terminal domains in the α3 chain removed by furin processing are not shown, although this step occurs after microfibril formation. Triple-helical monomers then form

disulfide-linked anti-parallel dimers with the approximate location of the disulphide bonds indicated by black arrows. Two dimers associate in a parallel fashion to form tetramers (for clarity, one dimer is shaded in grey). Tetramers are secreted into the extracellular space, where microfibrils are formed by the interlinking of the tetramers. Alternating tetramers are coloured grey for ease of visualisation. The double-bead and half-bead regions of the microfibril are highlighted by a black box (left and right, respectively). The bead region contains the same number of vWA domains as a tetramer, with the half-bead being equivalent to a dimer. Adapted from ref. 28; released under a CC BY 4.0 license [creativecommons.org].

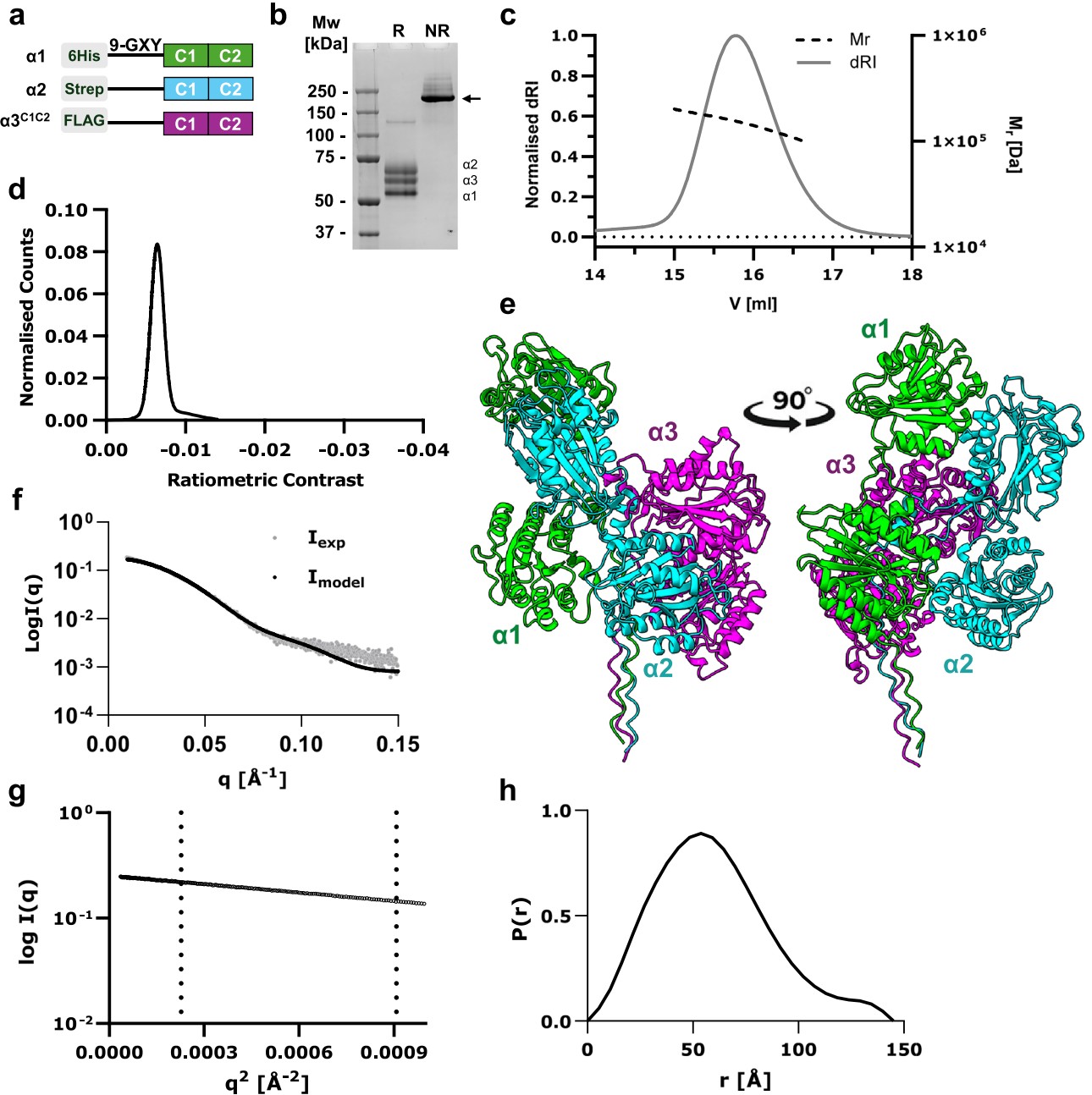

**Fig. 2 | Biophysical analysis of human mini-collagen VI α1α2α3$^{C1C2}$. a** Schematic representation of the mini-collagen VI α1α2α3$^{C1C2}$ constructs. These included N-terminal 6x His, Twin-Strep or FLAG-tags in the α1, α2 and α3 chains, respectively, followed by 9x Gly-X-Y triplets and the C1 and C2 vWA domains, with the furin cleavage site between the α3 chain C1–C2 domains mutated to prevent cleavage. **b** Coomassie-stained SDS-PAGE gel of the purified α1α2α3$^{C1C2}$ heterotrimer under reduced (R) and non-reduced (NR) conditions. The trimer dissociates into the three individual α-chains upon reduction of intermolecular disulphide bonds, as shown by three bands of the same intensity. Purifications were independently repeated at least 3 times with similar results. **c** SEC-Multi-angle light scattering of the purified α1α2α3$^{C1C2}$ heterotrimer indicated a molecular weight of 155.5 kDa, comparable to

its predicted molecular weight of 158.9 kDa. **d** Mass photometry also confirmed a single homogeneous species. **e** AlphaFold prediction of the α1α2α3$^{C1C2}$ heterotrimer is shown in cartoon representation in two orthogonal orientations. The α1, α2 and α3 chains are coloured green, cyan and magenta, respectively. **f** Experimental X-ray scattering data of the α1α2α3$^{C1C2}$ heterotrimer ($I_{exp}$, grey) plotted as a function of $q$, compared to the theoretical scattering of the AlphaFold model shown in black ($I_{model}$) with $\chi^2 = 1.76$. **g** The low $q$ scattering data of the α1α2α3$^{C1C2}$ heterotrimer is represented as a Guinier plot showing the linearity of the data across the Guinier region, as defined by dashed lines ($qR_g < 0.65$ and $qR_g \approx 1.3$). **h** Pair-distance distribution function $P(r)$ for the α1α2α3$^{C1C2}$ heterotrimer with maximum dimension ($D_{max}$) of 145 Å. Source data are provided as a Source data file.

sequential affinity purification steps, followed by size exclusion chromatography (SEC). Elutions from each purification step contained all three α-chains, indicating that heterotrimers had formed. A single band corresponding to the size of the heterotrimer was resolved on non-reducing SDS-PAGE, which upon reduction dissociated into the individual α-chains with sizes between ~52 and 70 kDa, indicating that

the heterotrimer is stabilised by interchain disulphide bonds (Fig. 2b). SEC-MALS analysis confirmed the presence of a single species with a molecular weight of 155.5 kDa (Fig. 2c) which was comparable to the expected molecular weight of 158.9 kDa for the heterotrimer, calculated from the amino acid sequences. Mass photometry also confirmed the presence of one homogeneous species (Fig. 2d).

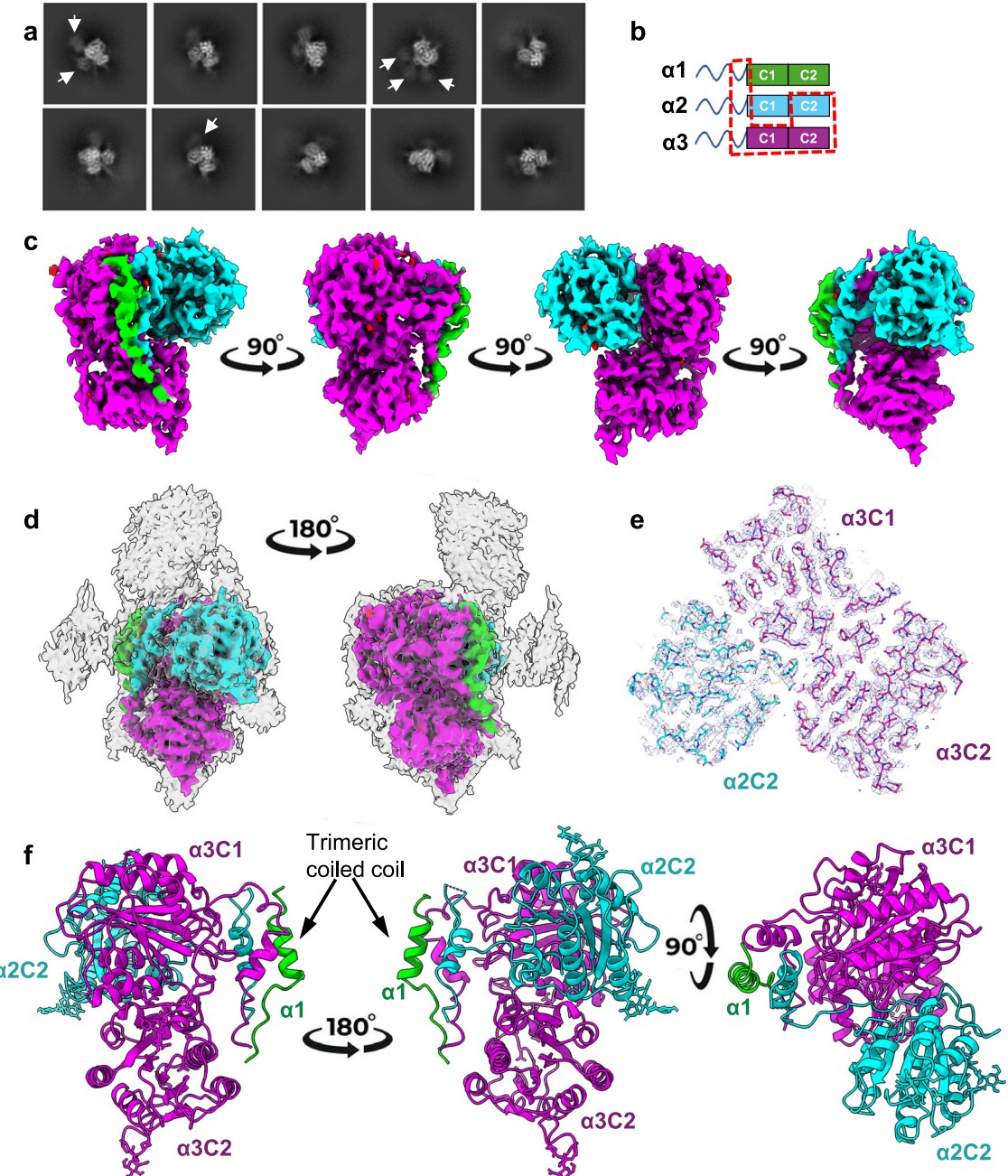

**Fig. 3 | CryoEM structure of the collagen VI α1α2α3$^{CIC2}$ heterotrimer.**
**a** Representative cryoEM class averages of the α1α2α3$^{CIC2}$ heterotrimer showing a core structure of three vWA domains. Diffuse density for three additional vWA domains was observed in some classes (white arrows). Box size is 260 × 260 Å. **b** Schematic diagram of the α1α2α3$^{CIC2}$ construct and highlighting with red dashed line three of the six vWA domains and a trimeric coiled coil preceding the collagenous region resolved in the final reconstruction. **c** 3.14 Å resolution density map after local refinement of the most structurally homogenous region shown in four orthogonal orientations. **d** The cryoEM density map at lower threshold level is superimposed and shown in grey in two orthogonal orientations. At this lower threshold, additional, diffuse areas of density likely to be the other three vWA domains can be observed. **e** Cross-section through the model with density map showing the α3 C1 and C2 domains and α2 C2 domain. **f** Cartoon representation of the structure of the α3 C1–C2 and α2 C2 domains and the coiled coil region is shown in different orientations. Maps and atomic models are coloured by chain (α1–green, α2–cyan and α3–magenta).

Alphafold Multimer was used to predict a model for the α1α2α3$^{CIC2}$ heterotrimer (Supplementary Fig. 2). The resulting model had a predicted Template Model (pTM) score of 0.63 and an interface pTM score of 0.59, suggesting a degree of uncertainty in the prediction. The model suggests that the C1 and C2 domains of the α3 chain and the C2 domain of the α1 chain pack against the collagenous triple-helical region. The C1 domains from the α1 and α2 chains are sitting above a region at the core of the trimer, formed from short helices or coiled coils from each of the three chains (Fig. 2e). In order to evaluate

the validity of the AlphaFold model and analyse the shape and hydrodynamic properties of the α1α2α3$^{CIC2}$ heterotrimer, BioSAXS data was collected (Fig. 2f). The analysis showed a maximum dimension ($D_{max}$) of 145 Å and radius of gyration ($R_g$) of 47.7 Å (Fig. 2g, h). The theoretical scattering data computed from the AlphaFold model were compared to the experimental SAXS data (Fig. 2f). The $\chi^2$ of 1.76 suggests a reasonable correlation with the experimental data, indicating that the AlphaFold model is a good representation of the α1α2α3$^{CIC2}$ heterotrimer at low resolution.

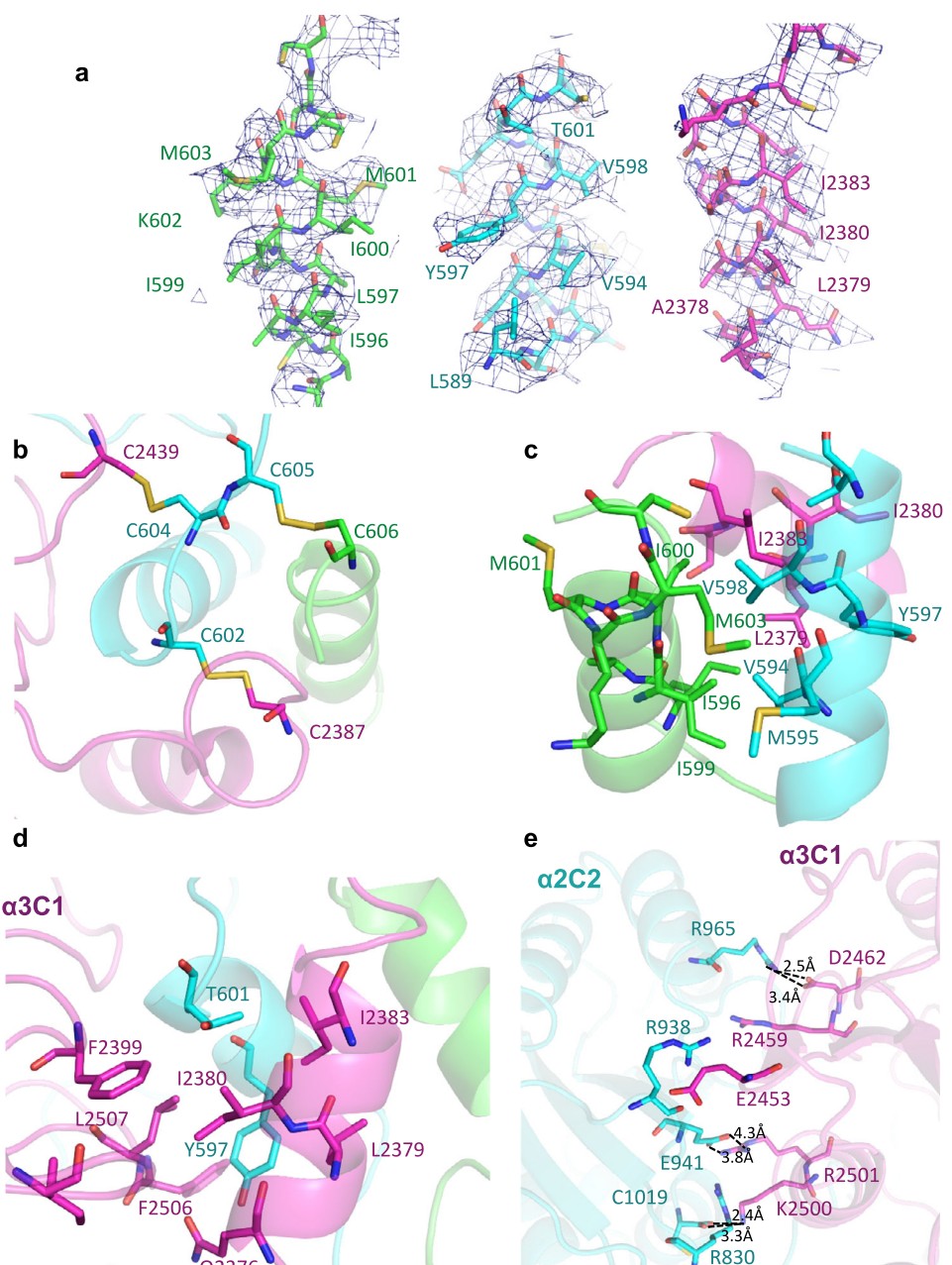

**Fig. 4 | Structure of the coiled coil region and interdomain interfaces. a** CryoEM density maps segmented for each chain, showing separately the α-helix in each chain that forms the coiled coil. Atomic models have been coloured by chain (α1–green, α2–cyan and α3–magenta). Where shown, non-carbon atoms are coloured according to the CPK convention. **b** View down the coiled coil showing the interchain disulphide bond pairing at the base of the coiled coil. **c** Side view of coiled coil region. **d** Intramolecular interactions between the α3 C1 domain and the coiled coil region, which stabilise this C1 domain. **e** Residues involved in the intermolecular interface between the α2 and α3 chains, involving their C2 and C1 domains respectively. A number of hydrogen bonds are involved in this interface.

## α1α2α3^{C1C2} heterotrimer structure and identification of a coiled coil

To determine the high-resolution structure, the α1α2α3^{C1C2} hetero-trimer was imaged using cryoEM and analysed using single particle analysis (SPA) (Supplementary Fig. 3). Processing of the cryoEM data revealed that there was substantial flexibility between the vWA domains, presumably due to the interdomain linkers, but a core of three vWA domains could be resolved, with additional diffuse areas of density, likely to be the other three vWA domains, seen in the 2D classes (Fig. 3a). Local refinements of the most structurally homogenous region yielded a 3.14 Å resolution structure (Fig. 3c; Supplementary Fig. 4), with additional diffuse areas of density for the other three vWA domains seen at higher thresholds (Fig. 3d). An AlphaFold

Multimer prediction for α1α2α3^{C1C2} was used to generate models of the individual vWA domains for rigid body docking, where the resolution of the map allowed unambiguous identification of the domains from sidechain densities. The AlphaFold model was broadly accurate, but the solution of the experimental model still required the manual rebuilding of various poorly predicted areas prior to refinement. Chain assignment was further cross-validated for the α1α2α3^{C1C2} model through comparison with the bovine microfibril reconstructions (explained in further detail in the methods). The three vWA domains, which were resolved to high resolution, were the α3 C1 and C2 domains and the α2 C2 domain (Fig. 3b, e), where the α3 C2 domain is adjacent to the α3 C1 domain. A trimeric coiled coil region, also predicted in AlphaFold, could be seen, and emerging from the coiled coil, a short

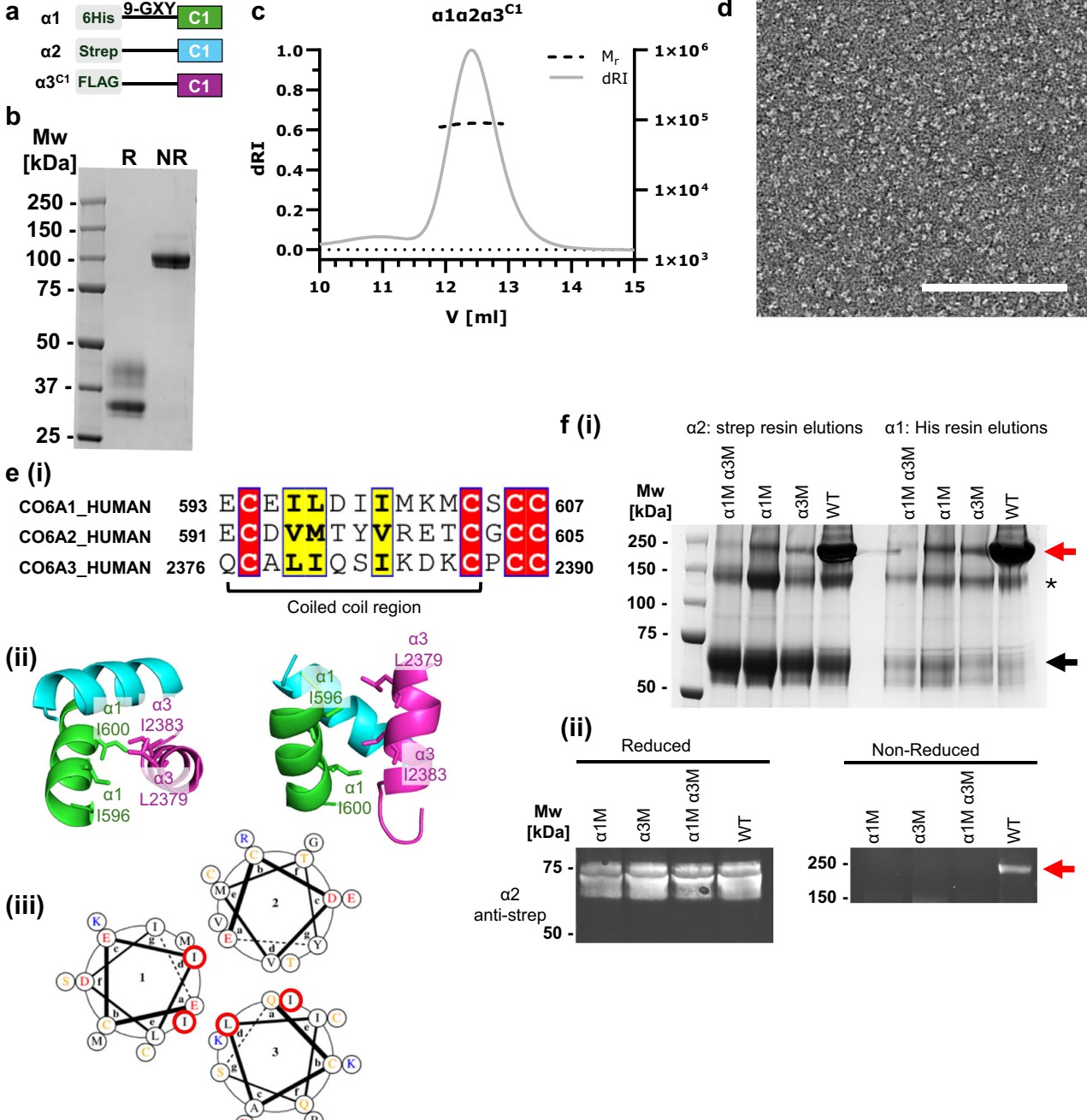

**Fig. 5 | Role of the C-terminal domains and coiled coil in trimerization.**
**a** Schematic of the mini-collagen VI α1α2α3$^{C1}$ constructs, which included
N-terminal 6x His, Twin-Strep or FLAG-tags in the α1, α2 and α3 chains respectively,
followed by 9x Gly-X-Y triplets and the C1 domains. **b** Coomassie-stained SDS-PAGE
of the purified α1α2α3$^{C1}$ heterotrimer under reduced (R) and non-reduced (NR)
conditions. The trimer dissociates upon reduction of intermolecular disulphide
bonds. **c** SEC-Multi-angle light scattering of the purified α1α2α3$^{C1}$ heterotrimer
determined a molecular weight of ~88 kDa, comparable to the predicted molecular
weight of 92.9 kDa. Source data are provided as a Source data file. **d** Negative-stain
EM image of the heterotrimers shows trimeric particles. Scale bar = 100 nm. **e**(i)
Sequence of the cysteine-rich coiled coil region from the α1, α2 and α3 chains of
human collagen VI. Cysteines are highlighted in red and conserved hydrophobic
residues in yellow. (ii) Structure of the mini-collagen coiled coil region with Ile and
Leu residues from the α1 and α3 chains forming a hydrophobic interface. (iii)

Helical wheel representation of the residues in the coiled coil, coloured orange
cysteine, red negatively charged, blue positively charged and grey hydrophobic.
Mutations were introduced into residues highlighted with red circles (α1 Ile596,
Ile600 and α3 Leu2379, Ile2383). **f** (i) Non-reduced SDS-PAGE of conditioned media
from wildtype and mutant α1α2α3$^{C1C2}$ (α1M (I596E, I600E) and/or α3M (L2379E,
I2383E)) following pull downs with the Twin-strep (α2) then His-tag (α1). A red
arrow indicates the disulphide-bonded heterotrimer and the monomeric chains
with a black arrow. Some dimers form between chains, indicated by an asterisk. (ii)
Western blot using an anti-Strep antibody of conditioned media from wildtype or
mutant α1α2α3$^{C1C2}$ in reduced and non-reduced conditions. A band for the het-
erotrimer was only observed for the wildtype but expression of individual chains
was unaffected. All purifications and blots were independently repeated at least 3
times with similar results.

stretch of collagenous region could be resolved, showing that the recombinant mini-collagen VI had formed a triple-helical structure (Fig. 3f).

The trimeric coiled coil has hydrophobic amino acids involved in stabilising the internal surface (Fig. 4a, Supplementary Fig. 5). Several cysteine residues can be found in the coiled coil motif, three inter- and intrachain disulphide bonds form, which have been modelled into the structure based on the presence of density. Other disulphides may also be present, but were not modelled if there was insufficient density to do this confidently. The coiled coil is stabilised by two interchain disulphide bonds between α3 and α2, and α2 and α1 chains (Fig. 4b). These interchain bonds explain the band seen for the heterotrimer on non-reducing SDS-PAGE, as disulphides are cross-linking the three α-chains. The three domains are packed together, stabilised by salt bridges and hydrophobic interactions with the coiled coil. The α3 C1 domain packs closely against the coiled coil, with Tyr597 from the α2 chain coiled coil contributing to a hydrophobic pocket with residues Ile2380, Phe2399 and Phe2506, which likely stabilises the interface (Fig. 4d). The opposite side of the α3 C1 domain interacts with the α2 C2 domain, where salt bridges are formed between residues Asp2462 and Arg965 and Arg 2501 and Glu941 (Fig. 4e). Density was present for four N-linked glycans in the cryoEM map, two on each of the α2 and α3 chains (Supplementary Fig. 6).

## Heterotrimerisation of collagen VI mediated by the coiled coil region

Collagen VI has a complex multistep assembly as shown in Fig. 1b and it is not fully understood which regions are important for trimerisation and triple-helix formation. Generally, in the assembly of fibrillar collagen, the C-terminal non-collagenous domains typically nucleate trimerisation. To determine whether the C2 vWA domains are important in this initial assembly step, a mini-collagen VI construct truncated after the C1 vWA domain, α1α2α3$^{C1}$ was expressed (Fig. 5a). Following purification, there is a single band at ~100 kDa on a non-reducing SDS-PAGE, which is similar to the expected size for a heterotrimer (92.9 kDa). Under reducing conditions, this band is absent showing that α1α2α3$^{C1}$ heterotrimer is also stablised by interchain disulphide bonds (Fig. 5b). The size of the heterotrimer was confirmed with SEC-MALS analysis, which determined a molecular mass of ~88 kDa and the trimer had a globular appearance similar to the α1α2α3$^{C1C2}$ heterotrimer when imaged by negative-stain TEM, showing that the C2 domains are not required for heterotrimer formation (Fig. 5c, d). Unfortunately, it was not possible to resolve a high-resolution structure of the α1α2α3$^{C1}$ heterotrimer by cryoEM due to conformational flexibility, which is consistent with the behaviour of the α1 and α2 chain C1 domains in the α1α2α3$^{C1C2}$ structure.

To determine the contribution of the coiled coil region in trimerisation of collagen VI, mutations were introduced in the α1 and α3 chains to residues on the internal surface of the coiled coil with the aim of perturbing this region. Helical wheel plots show that a hydrophobic interface is formed from residues in the 'd' and 'a' positions of the coiled coil, so mutations I596E, I600E in the α1 chain and L2379E, I2383E in the α3 chain were introduced (Fig. 5e). The mutant α1 and α3 chains were co-transfected with the α2 chain in Expi293F cells. The mutations did not affect expression of the individual chains but dramatically reduced the amount of heterotrimer formed when either mutant α-chain was expressed (Fig. 5f). This disruption to the formation of the heterotrimer shows that the coiled coil is important for trimer formation and that the C-terminal vWA domains are not essential requirements for trimerisation.

## Collagen VI microfibril double-bead structure from mammalian tissue

To determine how collagen VI is organised, microfibrils from bovine cornea, which we have previously shown are mainly composed of α1,

α2, α3 heterotrimers[27], were imaged using cryoEM. Microfibrils from bovine cornea are used due to their availability and abundance, with the bovine α-chain sequences around 90% identical to human. The double-bead and interbead region was analysed using single particle averaging to give a low-resolution structure (Supplementary Figs. 7 and 8), similar to previous observations[28,29]. To overcome flexibility in the microfibril bead and improve resolution, separate local masked refinements were carried out. One bead was masked and refined locally with C2 symmetry, which was subsequently relaxed using symmetry expansion. The bead structure was resolved to 6.36 Å resolution in its entirety (Fig. 6a; Supplementary Figs. 9 and 10), showing a layered structure of rings of vWA domains. At this resolution, the characteristic Rossmann fold of the vWA domains, a central β-sheet surrounded by 6 α-helices, can be resolved. The top of the single bead has a dense upper ring consisting of six vWA domains, below this ring are four vWA domains, then a further six vWA domains with lower resolution (Fig. 6a). In addition to the globular domains, two triple-helical collagenous structures can be seen running through the centre of the bead structure and another two collagenous regions can be seen emerging from the vWA domains at the top of the structure.

## Structure of the C-terminal region in the microfibril bead

At ~6 Å resolution, it is difficult to unambiguously identify each vWA domain in the microfibril reconstruction due to their similarity. Moreover, variability analysis showed that there was still a high degree of flexibility across the bead. Therefore, a further local masked refinement of the top ring of the bead, which has the highest resolution, was applied. Here, C2 symmetry was imposed, then relaxed using symmetry expansion to attain a nominal resolution of 4.33 Å in this region (Fig. 6b; Supplementary Figs. 11 and 12). Our previous modelling suggested that this upper ring of the microfibril contains the C-terminal vWA domains and the map also includes the trimeric coiled coil region, as seen in the mini-collagen VI structure, and a collagenous region from another collagen VI heterotrimer. Thus, aided by the mini-collagen VI structure and the AlphaFold2 prediction of the bovine C-terminal region, the α3 C1 and α2 C2 domains, the coiled coil and the α1 C2 domain were docked individually as rigid bodies, followed by manual rebuilding and real-space refinement. Supplementary Fig. 13 shows the overlay of the coiled coil region, α3 C1 and α2 C2 domains between the mini-collagen and bovine microfibril structures with an RMSD of 1.335 Å. Extracellular furin, MMP and BMP1 proteases remove the C2-C5 domains after microfibril assembly, so the α3 C2 domain is not present in the mature microfibrils[12,30]. The symmetry pair was then generated using non-crystallographic symmetry and the full model was refined.

The structure shows that the trimeric coiled coil region, identified in the mini-collagen VI structure, is also present in the intact tissue microfibril with a network of hydrophobic residues in the internal coiled coil (Fig. 6c). The α3 C1 domain is stabilised by interactions with the α2 coiled coil region, in a similar manner observed in the mini-collagen VI (Figs. 4d and 6d). The α1 C2 domain, which was mobile in the mini-collagen VI, is present in this structure and packs against the α2 C2 domain from the symmetry-related partner. This interaction is only formed upon higher-order assembly of the collagen VI heterotrimers into tetramers. Two interchain disulphide bonds are observed at the end of the coiled coil between the α3 chain and the α1 and α2 chains (Fig. 6e). Two collagenous regions are observed in the map, which are contributed by another collagen VI heterotrimer (Fig. 6b, f). This arrangement is formed upon dimerisation of the collagen VI heterotrimers, which pack in a head-to-tail arrangement (Fig. 1b). A cysteine residue from the α2 coiled coil region (Cys600) appears to be connected to and contact the collagenous region, suggesting the formation of a disulphide bond between the trimeric coiled coil region and collagenous region within the microfibril (Fig. 6f).

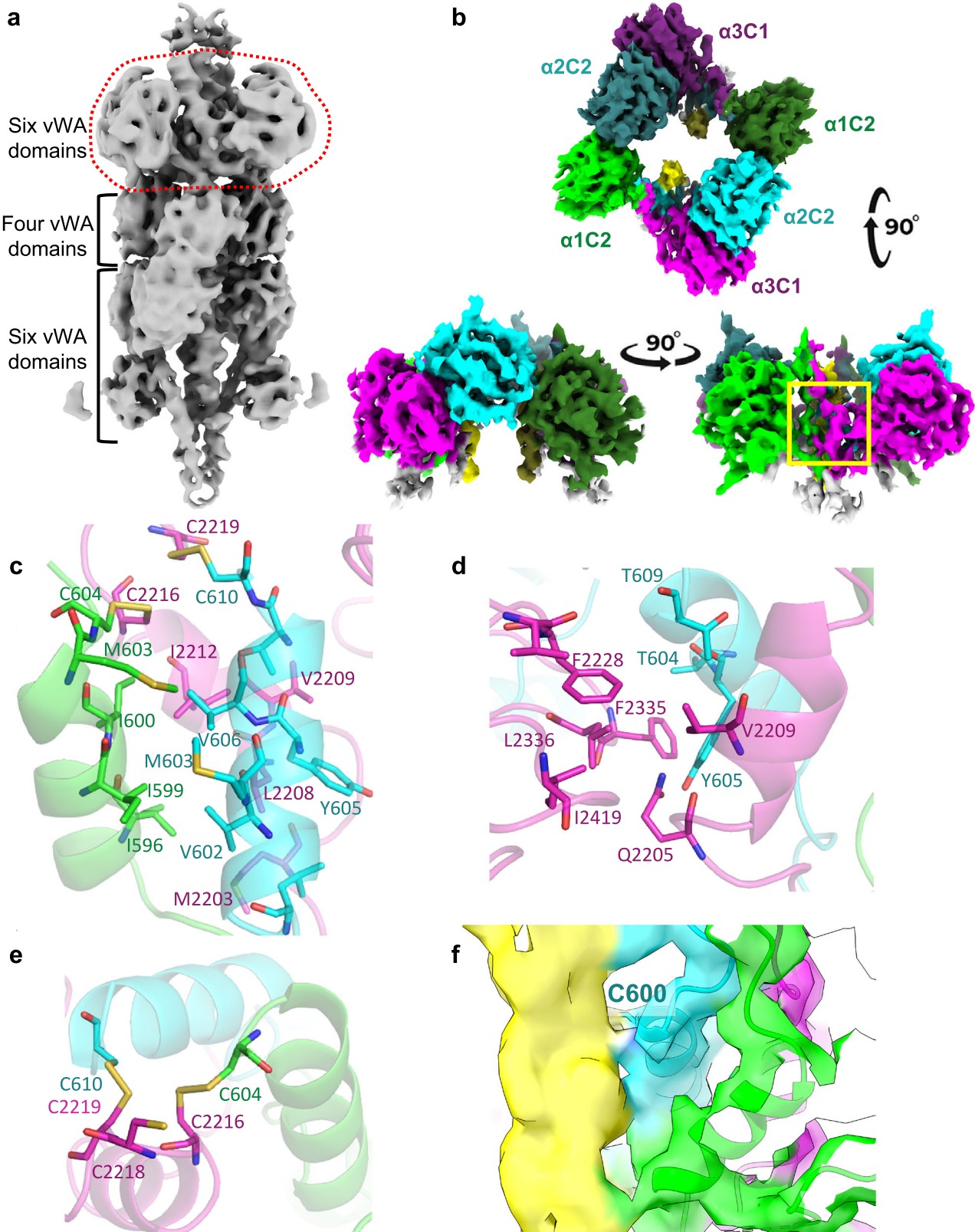

## Microfibril bead model reveals important regions in microfibril assembly

Having built the top ring of the microfibril, this allowed us to dock the α1 and α2 C1 domains into the layer below, where density connecting the C1 and C2 domains from both chains was present (Fig. 7a(i)). The docking of these domains was also consistent with their relative positions in the AlphaFold prediction (Supplementary Figs. 14 and 15).

This resulted in the complete C-terminal region being built into the top two rings of the microfibril bead map. Running through the C-terminal region is a long stretch of the collagenous region observed above, formed upon dimerisation of the collagen VI heterotrimers (Fig. 7a(ii)). Based on earlier biochemical studies and antibody-labelling[1,31,32], the remaining vWA domains at the bottom of the bead are likely to be the N1 domains from the three α-chains. An AlphaFold model of the N1

**Fig. 6 | Structure of the C-terminal region within the tissue microfibril. a** 6.36 Å resolution cryoEM density map of a single bead from the double-bead region of bovine collagen VI microfibrils, which is composed of 16 vWA domains in three layers. Highlighted in red is the region that was further locally refined in (**b**). **b** 4.33 Å resolution cryoEM density map of the C-terminal region of bovine collagen VI microfibrils. The map is shown in three orthogonal views, and in one view, the coiled coil region is highlighted with a yellow box. **c** A side view of this coiled coil region. **d** The intramolecular interactions between the α3 C1 domain and the α2 coiled coil region are shown. **e** View down the coiled coil showing the interchain disulphide bond pairing at the base of the coiled coil. **f** Side view of the coiled coil region in the density map, showing residue Cys600 from the α2 chain in density, which connects to the collagenous region. For (**b**–**f**), maps, cartoons or atomic models have been coloured by chain (α1−green, α2−cyan and α3−magenta) and the collagenous regions running through the middle coloured yellow. Where shown, non-carbon atoms are coloured according to the CPK convention. The symmetry-related domains from a second heterotrimer are shown in a darker shade.

domains from all three chains, was docked into the bottom of the bead, which correlated well with the cryoEM density (Supplementary Fig. 15). The model fitted in only one orientation, with the α1 and α2 N1 domains above and the α3 N1 domain underneath (Fig. 7b), which placed the beginning of the triple-helical region in the appropriate density. When docked in this orientation, at higher threshold values, there is extra density connected to the α3 N1 domain, likely the flexible α3 chain N2-N9 domains[25,27,33]. However, in the absence of a high-resolution map for this region, we cannot exclude the possibility that the N1 domains from the three chains could be assigned differently in this region, but this would not substantially change the modelling described below.

With the N-terminal domains docked, it was possible to model the collagenous region traversing through the bead region. The collagenous regions of the three α-chains each contain a single cysteine residue at positions 345(α1), 352(α2) and 1916(α3), which have been suggested to be important in dimerisation. The distance from the N-terminal domains of one tetramer to the C-terminal domains of the neighbouring tetramer is 26.3 nm, which corresponds to ~90 residues as the repeating Gly-X-Y motif has a helical rise of ~2.6 residues per nm. Based on the distance from the N-terminal domains, the approximate positions of these cysteine residues could be located. Cys345(α1) and Cys352(α2), both in the collagenous region, sit adjacent to the coiled coil region, in an area of density where the coiled coil region contacts the collagenous region. This suggests that a disulphide bond forms between either Cys345(α1) or Cys352(α2), in the collagenous region, with Cys600(α2) from the coiled coil region (Figs. 6f and 7c). Cys1916(α3), also in the collagenous region, is positioned at the bottom of the bead in the region where the two collagenous regions within the tetramer contact each other. Thus, an interchain disulphide bond could form between two parallel Cys1916(α3) residues, covalently linking and stabilising the two dimers in the tetramer (Fig. 7c).

**Pathogenic mutations located within intermolecular interfaces**
Having built a model for the bead, the full double-bead in the microfibril could be generated. This model could be used to analyse the locations of pathogenic UCMD and Bethlem myopathy mutations from the Human Gene Mutation database (HGMD)[34]. In particular, mutation hotspots where a number of missense mutations, affecting glycine residues, cluster at the N-terminal end of the collagenous region in each α-chain[35,36]. The positions of these mutation hotspots were mapped onto the microfibril model, and were found in the central region of the double-bead (Fig. 8). This region is formed from two overlapping tetramers in microfibril assembly where four triple-helices coil around each other and the N-terminal vWA domains contact the collagenous region. This region will support interactions important for microfibril assembly, which would explain the prevalence of pathogenic mutations in this region.

Conserved hydrophobic residues form the core of the trimeric coiled coil (Fig. 5). In the α1 and α3 chains, there are Ile and Leu residues in the a and d positions in the coiled coil, whereas in the α2 chain they are smaller valine residues (Fig. 8c). Interestingly, there are two variants that can occur in these residues (V594I and V598M), that may be pathogenic[37,38], which could be consistent with larger hydrophobic residues in the α2 chain perturbing the internal packing within the

trimeric coiled coil. Variants D835E and D905Y in the (α1)C2 domain correlate with the risk of spontaneous Coronary Artery Dissection (CAD)[10]. This region of the C2 domain is in a tetramer interface within the bead thus may be important in forming or stabilising tetramer-tetramer interactions within the microfibril (Fig. 8d). The (α2) chain C-terminal domains contain a number of pathogenic variants, three of which mapped onto interfaces between the (α2)C2 domain and the (α3)C1 domain within the heterotrimer and the (α1)C2 domain within the tetramer interface (Fig. 8e).

## Discussion
Here, we present the cryoEM structure of the bead region of the collagen VI microfibril purified from mammalian tissue. Extracellular matrix fibrils are resistant to structural biology approaches, as they cannot be made in vitro because of their complex hierarchical assembly. Therefore, we utilised a hybrid approach to determine the structure of a resolvable C-terminal region in the collagen VI microfibril, aided by mini-collagen VI constructs and AlphaFold predictions. Local masked refinement of the mini-collagen VI constructs and the equivalent region of the collagen VI microfibril generated structures of 3.14 Å and 4.33 Å resolution, respectively. Once this C-terminal region was built into the microfibril map, it was then possible to dock the remaining vWA domains into the 6.36 Å full bead map along with the collagenous triple-helical region.

Our structures of the C-terminal region of collagen VI, both from the recombinant mini-collagen VI and from tissue microfibrils, show the presence of a trimeric coiled coil region preceding the collagenous region, which appears important for the formation of a heterotrimer. Such trimeric coiled coils have been observed in fibrillar collagens (collagen I, III)[26,39], and have been suggested, based on sequence analysis, to support a ubiquitous role in collagen trimerisation[40]. However, for collagen VI, there was no evidence that this was the case, and it was not clear whether the vWA domains supported trimerisation. There is a lack of evidence in our structures that the vWA domains form trimeric units. Instead, the C1 domains appear to be mobile in the collagen VI heterotrimer, such that the α1 and α2 chain C1 domains are not visible in the EM map. Rather, the trimeric coiled coil region appears to be important for hetero-trimerisation. Previously, this coiled coil motif could not be predicted based on sequence alone[40], but AlphaFold was able to predict this region, which could then be experimentally confirmed. Inspection of the coiled coil region in our cryoEM structures showed a network of hydrophobic residues on the internal face. Hydrophobic residues are important driving forces for the supercoiling of α-helices, and hydrophobic residues (like leucine, valine, or isoleucine) are often found in the a and d positions in the heptad repeat[41]. Inspection of our model and helical wheel predictions showed that in these positions were valine, isoleucine and leucine residues. To test the importance of the coiled coil in the formation of the heterotrimer, mutations were introduced changing conserved residues, Ile596 or Ile600 in the α1 chain or Leu2379 or Ile2383 in the α3 chain (in the d and a positions in the heptad repeat) to glutamate. Single or double point mutations dramatically reduced the formation of the hetero-trimer, showing that this region is important in the assembly and stabilisation of the collagen VI heterotrimer.

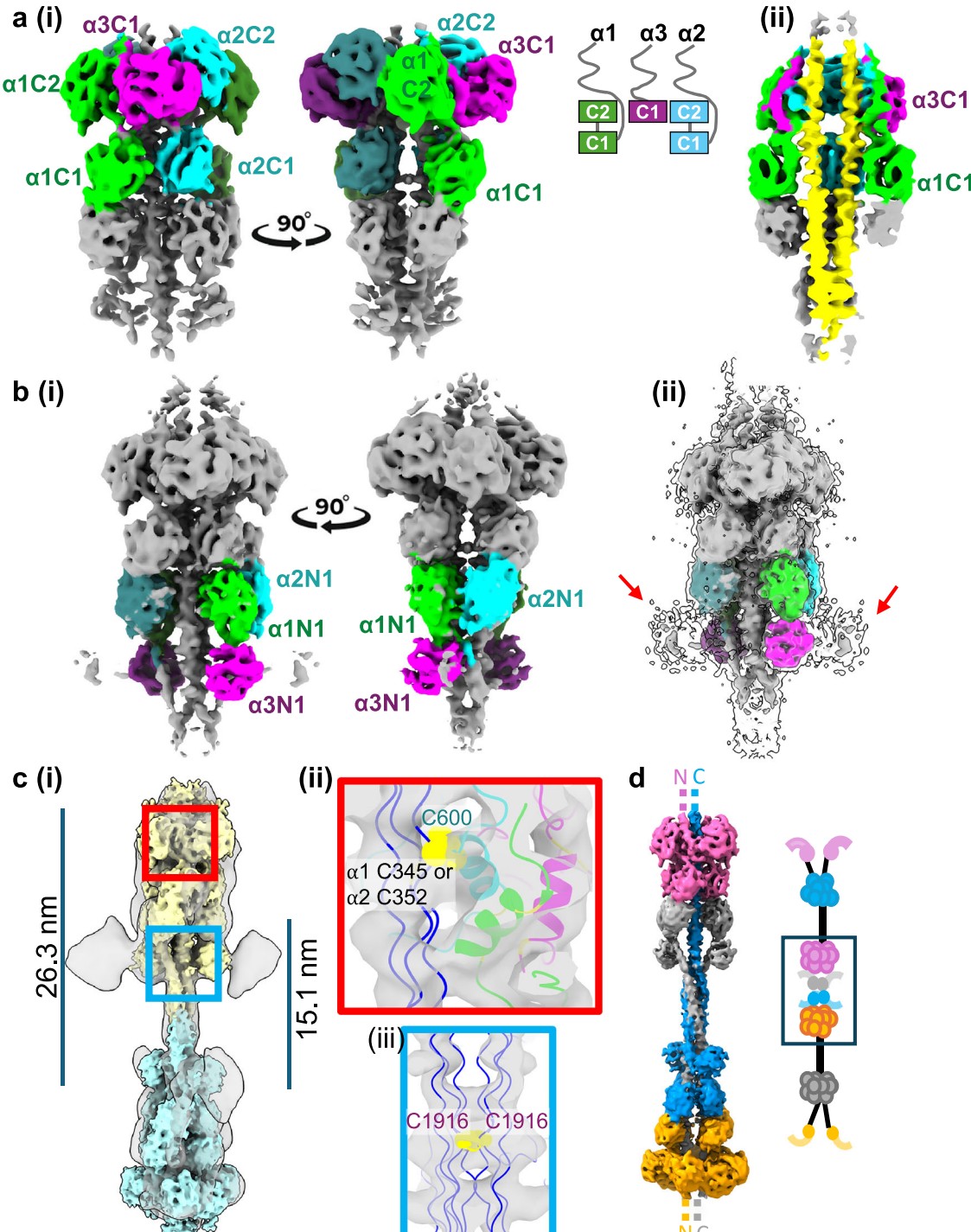

**Fig. 7 | Structure and organisation of the collagen VI tissue microfibril. a**(i) The EM map of the bead is shown in two orientations with the α1 and α2C1 domains docked with the C-terminal region from Fig. 6b. The schematic shows the arrangement of the C-terminal domains in layers in the bead. (ii) A cut through of the bead with the collagenous region, shown in yellow, running through the centre. **b**(i) The N1 domains from the three α-chains docked into the N-terminal region of the bead, shown in two orientations. (ii) Red arrows point to an area of extra density seen at lower threshold values, which is likely the additional flexible N-terminal domains from the α3 chain. In (**a**, **b**), maps have been coloured by chain (α1−green, α2−cyan and α3−magenta). **c**(i) Two single-bead cryoEM density maps (coloured in yellow and blue) are shown docked into the double-bead cryoEM density map. The red box highlights the microfibril coiled coil region and the blue box a region where the two collagenous regions connect. (ii) The area in the red box is enlarged and shown with the atomic model of the coiled coil region and an AlphaFold prediction of the collagenous region. Cys600 from the α2 coiled coil and α1 Cys345 or α2 Cys352 are shown as yellow spheres. (iii) The area in the blue box is enlarged where the two collagenous regions connect. Residues Cys1916 from two α3 chains are highlighted as yellow spheres. The distances between the N-terminus of the collagenous region to where the two collagenous regions connect and to the coiled coil are 15.1 nm and 26.3 nm, respectively. **d** A composite map of the double-bead with the N- and C-terminal regions coloured by overlapping tetrameric units as illustrated in the schematic, which shows how the C- and N-termini of adjacent tetramers overlap in an assembled microfibril.

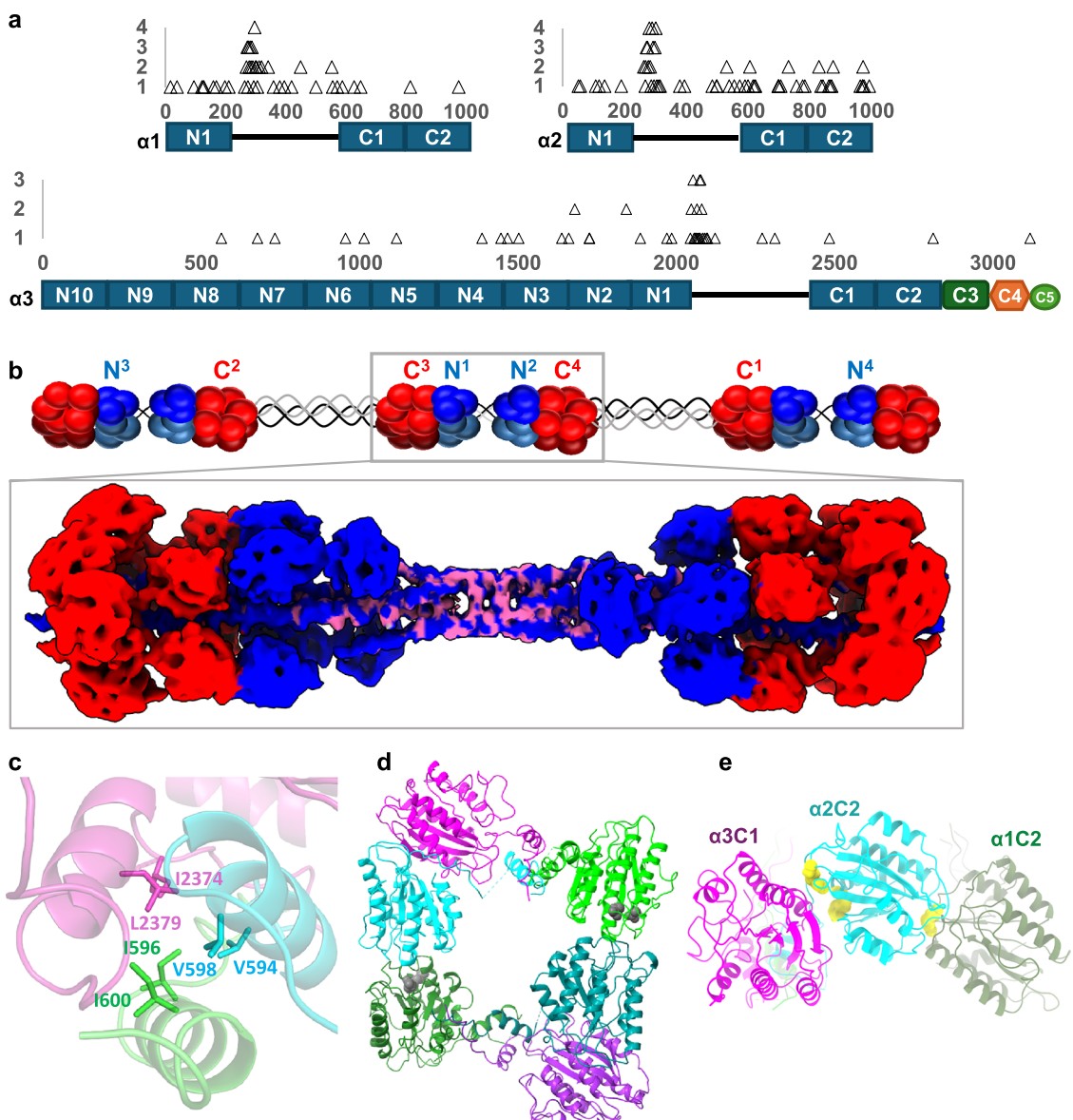

**Fig. 8 | Mapping UCMD and Bethlem myopathy pathogenic mutations on the microfibril structure. a** Frequency plots of missense variants from the HGMD database reported to be disease-causing in collagen VI α1, α2 and α3 chains. The frequency of occurrence of variants on any given amino acid is shown above the domain schematics of collagen VI, approximately to scale, showing in which domains the variants occur. The number of triangles on the *y*-axis corresponds to the frequency of the occurrence of a mutation on any amino acid. There is a hotspot of Glycine mutations at the N-terminal end of the collagenous region in each of the α-chains[35,36]. **b** A composite density map of the collagen VI double bead is shown, coloured blue for the N-terminal domains and red for the C-terminal domains. The high-frequency Glycine mutations in the collagenous region are coloured in pink, which fall in the collagenous region that traverses the bead region. Above is a model of the microfibril, with connecting N- and C-terminal regions numbered, showing that the bead is compiled from multiple collagen VI monomers. **c** The hydrophobic residues in the core of the trimeric coiled coil region show residues α2 V594 and V598, where variants that may be pathogenic can occur. **d** Variants D835E and D905Y in the (α1)C2 domain, which correlate with the risk of spontaneous Coronary Artery Dissection, are shown in grey in a tetramer interface within the bead. **e** Three Bethlem myopathy pathogenic variants in the (α2)C2 domain[71,72] localise to the domain interfaces between the (α3)C1 (R830W and R843W) and (α1)C2 (W1017R) domains, shown as yellow spheres. **c–e** are coloured by chain (α1–green, α2–cyan and α3–magenta), with symmetry-related chains shown in darker shades.

This trimerisation motif differs from the assembly of fibrillar collagens (collagen I, III, V, XI), where current models suggest that the C-terminal ColFi (NC1) domains in these collagens play a critical role in chain selection and trimerisation. These NC1 domains facilitate the correct alignment and association of procollagen chains, ensuring the formation of the appropriate heterotrimer[26,39], and a discontinuous 15-amino acid sequence within the NC1 domain is sufficient to drive the formation of protease-resistant triple helices[42]. Fibrillar collagen NC1 domains also have a conserved stretch of four cysteine residues that form a cysteine knot and when one of these cysteines is missing, some assemblies cannot form[43]. However, a recent paper shows that the C-terminal sequence of the triple-helical domain is also important in ultimately dictating what heterotrimeric assemblies are finally formed[44]. In the FACIT collagen IX, other non-collagenous domains are instrumental in chain selection and heterotrimerisation. The NC2 region of type IX collagen promotes assembly of specific chain combinations and defines the chain register of the collagenous region, where this NC2 region is coiled coil in nature[45].

In collagen VI, the trimeric coiled coil region is adjacent to a cysteine-rich sequence, previously termed a cysteine-rich junctional

area between the Gly-X-Y region and C-terminus, and suggested to play a role in disulphide bond formation to stabilise the triple-helical region[46]. A conserved CXC motif in the trimeric coiled coil region found in all three α-chains is involved in the formation of interchain disulphide bonds that stabilise the heterotrimer in both the mini-collagen VI and tissue microfibrils. Although the human α1α2α3$^{C1C2}$ reconstruction has near-atomic features, the map does show some level of flexibility. As a result, only three intermolecular disulphides could be modelled with certainty between the α1 and α2 chain, and two additional disulphide bonds between the α2 and α3 chains. The most clearly defined disulphide bond links α2 Cys602 to α3 Cys2387, but some uninterpretable density is visible around α3 Cys2390, suggesting other conformations may exist. All these cysteines are found in the CXC motif in the trimeric coiled coil region, except for one cysteine, which is ~50 residues downstream in the α3 C1 domain. In the tissue microfibril reconstruction, not all of these residues could be resolved in the map, but one bond between the α2 and α3 chains is visible, as well as another between the α1 and α3 chains. There is one difference in disulphide bond pairing in the CXC motif in the coiled coil region between the recombinant mini-collagen VI and microfibril, where in the mini-collagen VI, residue Cys2387 α3 bonds with Cys602 in the α2 chain, whereas in the microfibril, the equivalent cysteine in the α3 chain bonds with Cys604 in the α1 chain. It is possible that some cysteines may be unpaired or there may be heterogeneity in disulphide bond pairing, with nine cysteine residues found in close proximity. In the α1α2α3$^{C1C2}$ map, it is plausible that α1 Cys604 could pair with α3 Cys2387 and α2 Cys602 might pair with α3 Cys2390, as seen in the microfibril, which is more likely to represent the physiologically relevant pairing. The latter pair may be present at low occupancy in the recombinant sample, but the limited efficacy of heterogeneous refinement in a small asymmetric complex prevented the extraction of this discrete state. Chaperone proteins such as Hsp47 and protein disulphide isomerase (PDI) involved in post-translational modifications may assist in the folding and assembly of heterotrimers. Indeed, Hsp47 is important in the assembly of collagen VI[47], and collagen VI is a PDI substrate and PDI is found in tissue microfibril extracts[48,49].

Within the collagen field, there is a conundrum of how the transition occurs from an α-helical trimeric coiled coil, with no vertical stagger, to a collagen triple-helix with a one-residue stagger. The stagger defines which chain is leading or trailing, so a α1α2α3 arrangement would not be equivalent to a α2α1α3 arrangement, for example. The mini-collagen VI structure shows how this occurs for collagen VI, as there is a short region of the collagenous triple-helical domain visible in the structure emerging from the coiled coil region. The interchain hydrogen bonds and stagger suggest that the α3 chain is trailing, the α2 chain is the middle and the α1 chain is leading, going from a C- to N-terminal direction using the convention defined in ref. 50.

Early biochemical studies suggested that the anti-parallel interaction of collagen VI heterotrimeric monomers in the microfibril was stabilised by disulphide bonds[1,46]. There is only one cysteine residue in the collagenous region of each chain, at positions 345/344 in the human (and positions 345 and 352 in the bovine) α1 and α2 chains. In our microfibril structure, these two cysteines in the α1 and α2 chains are adjacent to the coiled coil region of the anti-parallel monomer. Furthermore, there is density in the cryoEM map, which supports a bond forming between the collagenous and coiled coil regions. Indeed, Bethlem myopathy mutations that skip exon 14 in the α1 chain, deleting cys345, prevent dimer formation[51], which suggests this cysteine in the α1 chain collagenous region is bonding to the coiled coil region (Fig. 7c(ii)). These data suggest that the coiled coil region has a further important role and may also be involved in the formation of disulphide bonds stabilising the supramolecular dimerisation of collagen VI monomers (Fig. 1). The cysteine residue in

the collagenous region of the α3 chain is found at position 2087 (or 1916 in the bovine sequence), 30 residues into the collagenous region, which is closer to the centre of the double-bead. In this location, the two parallel strands of collagenous region from the adjacent beads are connected by density, suggesting that the tetramer is also stabilised by a disulphide bond. These observations fit well with previous suggestions that a minimum of two cysteine residues would be essential for oligomer cross-linking and would be located about 15 and 30 nm away from one end of the triple-helix[1].

Pathogenic variants in the α1, α2 and α3 chains cause collagen VI-related muscular dystrophy, which can act in a dominant negative manner. The majority of variants are found in the triple-helical region and affect glycine residues[36,52]. Analysis of the frequency of variants at a particular residue shows that there is a cluster of mutations occurring at the N-terminal end of the collagenous region for all three chains. As collagen assembles from a C-to-N-terminal direction, it was not clear why mutations at the N-terminal end of the collagenous region would be so frequently observed. Mapping these pathogenic variants onto the model of the microfibril double-bead shows that they cluster in the core of the bead. A previous study has shown that mutations in glycine residues at the N-terminal end of the triple-helix compromised intracellular assembly and disulphide bonding of tetramers. Dimers containing mutant chains were able to assemble into tetramers, but stabilisation of the tetramers by disulphide bonding was compromised, and microfibril formation was impaired[36]. This suggests that these mutations can impair head-to-tail dimerisation and formation of the disulphide bond that stabilises the dimer, and also formation of higher-order assemblies. As these residues are in the interface formed when two collagen VI tetramers interact in microfibril assembly, it may be that interactions in this region of the triple-helix with the non-collagenous domains are essential for microfibril assembly. Furthermore, there is a UCMD variant, C602W[53], which changes the cysteine at one end of the α2 chain coiled coil region. Our structure shows that this residue in the CXC motif is involved in a disulphide bond linking the coiled coil regions of the α2-α3 chains, which would be lost upon mutation of this cysteine, and a bulky tryptophan residue may perturb the interaction of these domains in the heterotrimer.

In addition to pathogenic mutations, variants in collagen VI are increasingly being linked to poor disease outcomes, such as increased risk of cardiovascular disease, and recently, super-structures of amyloid and collagen VI were observed in heart tissue from an amyloidosis patient[29]. Although the pathological significance is unknown, two variants in COL6A1 have been found in patients with Spontaneous Coronary Artery Dissection[10], which include changes to aspartic acid residue, D835E in the C2 domain of the α1 chain. Some vWA domains, such as the I-domains in integrins, have MIDAS or metal-ion dependent adhesion sites, which have a non-contiguous sequence motif (D-x-S-x-S... T...D) that supports protein-protein interactions. Interestingly, the α1 C2 domain has a conserved MIDAS consensus sequence with D835, the first aspartic acid within this sequence. This region of the C2 domain is in an interface within the tetramer; thus, interactions mediated by this MIDAS sequence may be important in forming or stabilising tetramer-tetramer interactions within the microfibril. Given that a function of collagen VI is to maintain the mechanical strength of elastic tissues, such as major blood vessels, destabilising intramolecular interactions within the microfibril may offer an explanation why these variants increase the risk of dissection.

In summary, we have determined the structure of the extracellular matrix microfibrillar protein, collagen VI, which has revealed the steps in hierarchical collagen VI microfibril assembly. The structure has enabled us to identify a hotspot for mutations in the microfibril bead, in an important region for microfibril assembly. We expect the structure will be a valuable resource in understanding the location and impact of pathogenic mutations and variants.

## Methods

### Expression and purification of mini-collagen VI

Synthetic genes were obtained from ThermoFisher Scientific using the GeneArt Synthetic Gene service. Protein sequences for human collagen VI were obtained from UniProt ColVIα1$^{C1C2}$: P12109 (aa.566-1028); ColVIα2$^{C1C2}$: P12110 (aa.564-1019); ColVIα3$^{C1C2}$: P1211-1 (aa.2347-2820) and DNA sequences were generated using the GeneArt online portal and codon-optimised for *Homo sapiens*. All synthetic genes were cloned into pHLSec using AgeI and XhoI[54]. Proteins were transiently expressed in Expi293F suspension cells according to manufacturers' instructions, with 50 µg/ml L-ascorbic acid-2-phosphate (Merck) added with the enhancers 16–24 h after transfection. Cells were cultured for 5 days before conditioned media were collected, centrifuged at $4500 \times g$ at 4 °C for 15 min and filtered. The medium was supplemented with 2.5 mM *N*-Ethylmaleimide (NEM, Merck) and 18 ml/l BioLock blocking solution to remove naturally present biotin. To purify ColVIα1α2α3 trimers, a three-step affinity purification strategy using Strep-Tactin XT 4Flow High-Capacity resin (IBA LifeSciences) with protein eluted in 20 mM HEPES, 500 mM NaCl, pH 7.4, with 75 mM Biotin. Followed by immobilised metal-ion affinity purification (IMAC) using Ni-Sepharose™ Excel (Cytiva) with proteins eluted in 20 mM HEPES, 500 mM NaCl, pH 7.4, with 500 mM Imidazole. Anti-FLAG M2 Affinity Gel (Merck) was added to the elutions and proteins were eluted with 200 µg/ml 3× FLAG Peptide (Merck). SEC was used as the final purification step using either a Superdex 200 Increase GL column or Superose 6 increase 10/300 GL column on an Äkta Go FPLC system (Cytiva) at 0.5 ml/min in 20 mM HEPES, 150 mM NaCl, pH 7.4 (HBS). Protein elution was monitored using UV absorbance at 280 nm. Protein identity was confirmed using in-gel digestion and MS/MS by the University of Manchester BioMS Facility.

### Western Blotting

Western blotting was performed with anti-strep (IBA Lifesciences) primary antibody and anti-mouse Alexa Fluor secondary antibody and blots were imaged using a ChemiDoc™ MP Imaging System (BioRad).

### Mass photometry

Mass photometry was performed on a Refeyn TwoMP (Refeyn) at a final sample concentration of 0.1 µM. Data were collected for 60 s and processed in DiscoverMP 2.3 (Refeyn).

### Multi-angle light scattering (MALS)

Samples were loaded onto a Superdex 200 or Superose 6 column equilibrated with HBS at 0.75 mL/min. Eluted samples passed through a DAWN Wyatt Helios II 18-angle laser photometer with one of the detectors replaced with a Wyatt QELS detector coupled to a Wyatt Optilab rEX refractive index detector. The molecular mass and hydrodynamic radii of the resulting peaks were analysed using Astra 6.1 (Wyatt).

### AlphaFold protein structure prediction

AlphaFold2 Multimer Protein Structure predictions[55] were generated on the compute shared facility (University of Manchester) using AlphaFold version 2.1.1. Predicted models and their corresponding data files were transferred to a local machine, and heatmaps visualising the predicted alignment error were generated using a custom Python script deposited in the Zenodo database [https://doi.org/10.5281/zenodo.15880546]. AlphaFold 3 predictions used the AlphaFold server[56]. Models of human ColVIα1α2α3$^{C1C2}$, used the sequences described above, and the C-terminal model of bovine collagen VI was generated using UniProt sequences ColVIα1: E1BI98 (version 1), ColVIα2: F1MKG2 (version 2/A0AAA9TXG2) and ColVIα3: A0AAA9TAB4.

### Small-angle X-ray scattering (SAXS)

BioSAXS data were collected on Beamline B21 at Diamond Light Source (Oxford, UK). Data were collected on an EigerX 4M detector (Dectris, Baden, Switzerland) at a wavelength of 1 Å with a standard exposure time of 1.0 s and a $q$ range of 0.0045–0.34 Å$^{-1}$, where $q$ is defined as $q = 4\pi \frac{\sin(\theta)}{\lambda}$. Using the BioSAXS robot, 25 µl of sample or matched buffer was loaded into the capillary, and typically $20 \times 1.0$ s exposures were collected. The collected frames were averaged and buffer-subtracted using Diamond's Data Analysis Workbench (DAWN)[57] to generate 1-dimensional scattering data files for processing. SAXS data were processed using ScÅtter and the ATSAS software suite[58]. Data were analysed to calculate the radius of gyration ($R_g$) and produce the pair-distance-distribution function ($P(r)$) through inverse Fourier transform to determine the maximum dimension ($D_{max}$). Scattering profiles were calculated from the AlphaFold2 prediction (Supplementary Fig. 2) using the Fast X-ray Scattering[59] web service and the calculated $\chi^2$ was used to assess correlation. All graphs were plotted in GraphPad Prism and data collection parameters are shown in Supplementary Table 1.

### Collagen VI microfibril extraction

Bovine corneas were locally sourced from an abattoir, minced and digested overnight at room temperature with 0.1 mg/ml collagenase type 1a in digestion buffer (20 mM Tris HCL, 400 mM NaCl, 2.5 mM CaCl$_2$, 3 mM PMSF, 5 mM NEM) as previously described[27,28]. After digestion, samples were centrifuged and the supernatant separated using SEC using a Sephacryl S-500 column (Cytiva).

### Negative-stain transmission electron microscopy (TEM)

Protein samples were adsorbed to glow-discharged 300 mesh carbon-coated copper grids (Agar Scientific) and stained with 2% (w/v) uranyl acetate. Images were taken with 1 s exposure at 57,000× magnification on a Talos L120C TEM equipped with a Ceta 16M detector operating at an accelerating voltage of 120 kV.

### CryoEM sample preparation

Mini-ColVI samples were prepared in HBS or 20 mM Tris HCl, 150 mM NaCl, pH 7.4, at a concentration of 1.0–1.5 mg/ml and adsorbed to glow-discharged R2/2 holey carbon film on copper grids (Agar Scientific) and plunge-frozen using a VitroBot MkIV system (ThermoFisher Scientific). Purified collagen VI microfibrils were adsorbed onto grids using a multiple application approach[60]. Samples were adsorbed onto glow-discharged copper quantifoil r2/2 grids before being manually blotted to remove excess buffer. After multiple rounds of sample application and blotting, grids were blotted a final time for 4 s at blot force 0 and plunge-frozen in liquid ethane using a Vitrobot mk IV. CryoEM grids were screened using a Glacios 200 kV cryoEM. Images were collected on grids that had six sample applications and blotting steps.

### CryoEM Data collection and processing

**Mini-collagen VI α1α2α3$^{C1C2}$.** Data collection on the colVIα1α2α3$^{C1C2}$ sample was performed on a Titan Krios G2 cryoEM equipped with a Gatan K3 detector operating at an accelerating voltage of 300 kV at the Electron Bio-Imaging Centre (eBIC), UK. Movies were recorded at a magnification of 130,000x to give a pixel size of 0.65 Å/pixel and defocus range of −2 to −0.75 µm. Data acquisition was set with a dose rate of 12.94 e$^-$/pixel/s, a 1.09 s exposure giving a total dose of 28.11 e$^-$/Å$^2$ per movie, which were saved as non-gain corrected compressed TIFF format. The final dataset comprised 35,706 40-frame movies. Movies were imported into cryoSPARC[61], motion correction with dose weighting and CTF parameters were estimated using cryoSPARC live using the default parameters. Initial particles were picked using blob pick in cryoSPARC live using a maximum particle diameter of 100–150 Å. Blob picker picked 10,665,534 particles, which were initially extracted with a 300 ×300 pixel box, which was down-sampled to 64 × 64 pixels with a pixel size of 3.04 Å/pixel to speed up processing. This initial particle set was subjected to five rounds of 2D classification to remove bad particles. Each round, the best particles were selected

based on the appearance of their observable secondary structural elements and a resolution cut off of 8 Å. Bad particles, which consisted of ice contamination or classes with multiple overlapping particles, were removed. This resulted in a particle set of 20,210 particles, which were used to train a Topaz model[62], and generate three ab initio 3D classes. The trained topaz model was used to pick 4,288,503 particles, which were re-extracted with a box size of 300 pixels, which was downsized to a box size of 128 × 128 pixels with a pixel sampling of 1.52 Å/pixel. The Topaz picked particles were then subjected to three rounds of 2D classification and removal of bad particles, as previously described, which left a particle stack of 234,550 particles. The resulting particle stack was used to refine the ab initio volume, which came from the class with the largest number of particles using homogeneous refinement. The aligned particles were then re-extracted in a bigger 400 × 400 pixel box, which was down-sampled to a 200 × 200 pixel box with a pixel sampling of 1.3 Å/pixel. The re-extracted particles were then refined with homogeneous refinement, resulting in a reconstruction with a resolution of 4.14 Å.

The refined map was then 3D classified using heterogeneous refinement with the map from the last round of homogeneous refinement and the ab initio models as initial seeds for 3D classification. The 151,223 particles from the class with the highest resolution were refined using non-uniform refinement[63] and then local refinement. A binary mask focusing on the most structurally homogeneous region was generated by deleting defuse density using the map eraser in Chimera. The resultant map was then reimported into cryoSPARC and a binary map with a dilation radius of 2 pixels and soft padding of 12 pixels was generated using volume tools. Particles from the local refinement job were used to train a new Topaz model, which picked 6,491,410 particles. Particles were extracted in a 400 × 400 pixel box and were down-sampled to 200 × 200 pixels with a pixel size of 1.3 Å/pixel. Bad particles were removed as before, using two rounds of 2D classification to give a dataset of 438,007 particles. Particles were then 3D classified using heterogeneous refinement and the highest resolution class, containing 287,134 particles, underwent another round of non-uniform refinement and local refinement. More classes were selected from the previous round of 2D classification, giving a particle set of 1,011,536 and were subjected to two rounds of heterogeneous refinement. In both rounds, the 3D class with the highest resolution was selected, leaving a particle set of 378,487 particles. These particles were then used in a final heterogeneous refinement, where the locally refined reconstruction and a map with the poorest resolved density were removed using Chimera map eraser and were used as seeds for the 3D classification. The 246,984 particles from the best class were then used in a final local refinement, which resulted in a final map with a resolution of 3.14 Å (Supplementary Fig. 4). The data processing workflow is summarised in Supplementary Fig. 3 and cryoEM data collection parameters are shown in Supplementary Table 2.

**Bovine collagen VI microfibrils.** Data collection of bovine microfibril specimens was performed on the same microscope. Movies were collected in super resolution mode at binning 2 at a nominal magnification of 105,000×, which gave a pixel size of 0.829 Å/pixel, at a defocus range of −2 to −0.75 μm with an exposure time of 2.3 s at a dose rate of 15.234 e−/pixel/s, which gave a total dose of 50.98 e−/Å². A summary of cryoEM data collection parameters is shown in Supplementary Table 2. The final dataset comprised 29,595 50-frame non-gain compressed movies. Motion correction and dose weighting were performed using MotionCorr2[64] with 5 × 5 patches, in the eBiC ispyB pipeline. Motion corrected images were imported into RELION-4.0[65] where CTF parameters were estimated using CTFFind4[66]. 855 double-bead particles were manually picked and extracted in a 800 × 800 pixel box, which was down-sampled to 200 × 200 pixels, resulting in a 3.316 Å/pixel sampling. Particles were then aligned and classified using RELION 2D classification and the class averages with the characteristic

double-bead structure were selected, resulting in 325 good particles, which were used to train Topaz[62]. Topaz extraction picked 145,555 particles; these particle coordinates were imported into cryoSPARC[61]. One round of 2D classification in cryoSPARC was used to remove bad particles; 6930 particles from the best classes were selected and used to generate an ab initio model, which was then refined using homogeneous refinement. During 3D refinement, D2 symmetry was applied with the symmetry axes along and perpendicular to the fibre axis. Particles were then used for another round of Topaz training and Topaz extraction, which picked 809,113 particles. After particle picking, good particles were selected after three rounds of 2D classification and 154,056 particles were re-extracted in a 1200 × 1200 pixel box and down-sampled to 512 × 512 pixels with a pixel size of 1.94 Å/pixel. These particles were then used in a non-uniform refinement of the previous homogeneous refinement. To separate out the straightest particles, 2D classes of the straightest 34,301 particles, with the least flexibility along the fibre axis, were picked and refined separately in a non-uniform refinement. During non-uniform refinement, the symmetry was relaxed using the marginalisation option in the non-uniform refinement procedure. The reconstruction of the straightest particles and the previous non-uniform refinement were used as seeds for a final heterogeneous refinement of 154,056 particles from the previous 2D classification. The 47,488 particles from the straightest class from the heterogeneous refinement were then used in a final non-uniform refinement, during this refinement, D2 symmetry was relaxed again using marginalisation. The data processing workflow is summarised in Supplementary Fig. 7 and the final reconstruction of the single bead had a resolution of 12.3 Å (Supplementary Fig. 8).

Single-bead coordinates were selected from 2679 double-bead particles. Particles were aligned in a single class and the particles were centred and re-extracted at the centre of each single bead using RELION extract (box size of 512 × 512 pixels, down-sampled to 128 × 128 pixels with a 3.316 Å/pixel sampling). The two stacks of single beads were combined to form a single set of particles. The single beads were then 2D classified and 3,010 particles from the best single bead class averages were used to train Topaz to create a Topaz model for picking single beads. Topaz picked 1,434,373 single beads, which were subjected to six rounds of 2D classification to remove bad particles. The resulting 145,103 good particles were used to generate an ab initio model. This initial model was then refined using 3D auto refine, as previously reported, the bead has 2-fold symmetry along the fibre axis[27], so unless otherwise stated, all refinements were conducted with C2 symmetry along the fibre axis. The particles then underwent 3D classification; the class of particles, which gave the highest resolution reconstruction, contained 60,359 particles, which were then further 3D refined, before being used to train a new topaz model. Topaz picked 1,011,458 particles, which underwent 3 rounds of 3D classification to select particles from classes with the best resolution, resulting in a particle stack of 62,391 particles. These particles were then 3D refined again before being imported into cryoSPARC and used for non-uniform refinement[63]. Particles were then used for another round of Topaz training and Topaz extraction, which picked 1,891,914 particles. Bad particles were again removed using two rounds of 2D classifications to give a final dataset of 248,184 particles. Particles underwent another round of non-uniform refinement before particles were re-extracted in a 512 × 512 box and were down-sampled to 256 × 256 to give a pixel sampling of 1.65 Å/pixel and were used in a final non-uniform refinement. The data processing workflow is summarised in Supplementary Fig. 9 and the final reconstruction of the single bead had a resolution of 6.36 Å (Supplementary Fig. 10).

Due to flexibility along the bead axis, a tight mask was applied to the top layer of the single-bead and was used for local refinement. In this refinement, the symmetry was relaxed to C1 symmetry using C2 symmetry-expanded particles. To further assess heterogeneity in the data, 3D variability analysis was performed. Heterogeneous refinement

was used on particles using the best local refinement and a reconstruction of the highest resolution variability cluster as seeds for the classification. The 147,117 particles from the class with the highest resolution from heterogeneous refinement were then locally refined with no symmetry using C2 symmetry-expanded particles. These particles then had a final round of 3D classification and the best class, which contained 87,841 particles after duplicate particles had been removed. The particles from the 3D classification were again C2 symmetry expanded and were used for a final round of local refinement. The final locally refined top ring of the bead had a resolution of 4.33 Å (Supplementary Fig. 12) and the data processing workflow is summarised in Supplementary Fig. 11. The particles used in the refinement had a box size of 512 × 512 and were down-sampled to 256 × 256 to give a pixel sampling of 1.65 Å/pixel.

### Chain identification, model building and refinement

The globular vWA domains that comprise the majority of the domains in collagen VI are orthologous and contain a similar Rossmann fold, so it was important to determine the correct chain identity. AlphaFold2 multimer[55] or the AlphaFold 3 server[56] were used to generate models of heterotrimers (Supplementary Figs. 2, 16 and 17), from which the individual vWA domains, which had high confidence scores, were used as starting models for model building. Docking of whole models or individual domains and regions was performed using UCSF ChimeraX[67]. The AlphaFold models predicted the close interaction of two vWA domains (later confirmed to be (α3)C1 and (α2)C2), with this interacting pair of domains present in both the ColVIα1α2α3$^{ClC2}$ and bovine microfibril reconstructions. Once it was clear that these two domains were equivalent in the two reconstructions (regardless of their assignment by AlphaFold), it was possible to identify a domain which was present in the mini-collagen map but not present in the bovine map (Supplementary Fig. 14). This domain was the (α3)C2 domain, which is known to be removed by furin cleavage during microfibril maturation[30]. Chain tracing in the α1α2α3$^{ClC2}$ map, and analysis of the density at the C-terminus of the trimeric coiled coil region in both maps allowed the identity of the (α3)C1 domain to be confirmed, which is the vWA domain that interacts most closely with the coiled coil.

It was deduced that the third domain resolved in the α1α2α3$^{ClC2}$ map, which interacts with (α3)C1, was a C2 domain from either the α1 or α2 chain. This is because, in the microfibril reconstruction, each of the C1 domains is connected to the coiled coil region by two identifiable ascending and descending loops and one such descending loop connects with the resolved C2 domain in question. The C2 domains of collagen VI have a high degree of sequence similarity, especially in the hydrophobic core, but the reconstruction was sufficiently high resolution to allow direct residue identification from the experimental map. Structural alignment of models of the C2 domains allowed verification of the (α2)C2 domain, with residues Phe836, Pro971, Trp1017 and Met964, explaining the experimental density far better than the equivalent residues in the α1 or α3 chains. Additional validation of the vWA domains was obtained by inspecting the extreme C-terminal regions of the α1α2α3$^{ClC2}$ reconstructions, including the disulphide bond at Cys1019 in α2.

The relative direction of the coiled coil also required verification, and this was achieved first by identifying Tyr597 and Arg599 from the α2 chain, the former being the only aromatic residue in the coiled coil (Supplementary Fig. 5). The main-chain density linking the coiled coil with the (α3)C1 domain is strong enough to identify and model a contiguous chain, and the α1 chain, which is only resolved in the α1α2α3$^{ClC2}$ reconstruction as a short section of collagenous peptide and coiled coil, was placed by elimination. Several other features were used to verify the chain assignment in the coiled coil, including the sidechains of α1 Met603 and α2 Met595 and the visualisation of the aforementioned ascending and descending loops of the α2 chain. Once rigidly docked, initial models were obtained, manual rebuilding

was performed in COOT[68] and refinement was performed using Phenix real-space-refine[69] with validation using Molprobity[70]. Experimental maps and models were deposited to the EMDB and PDB, respectively. Validation metrics are shown in Supplementary Table 2.

For model building the remaining domains into the bead map, a model for a heterotrimer of the N1 domains with 100 amino acids of the collagenous sequence from the α1, α2 and α3 chains was predicted using AlphaFold 3 (Supplementary Fig. 18). This α1$^{N1}$α2$^{N1}$α3$^{N1}$ model was docked as a rigid body into the single bead map using ChimeraX[67].

### Mapping of collagen VI variants

Single-nucleotide variants from the HGMD database[34], where pathogenicity was indicated as disease-causing (DM) with UCMD or Bethlem myopathy phenotype, that resulted in an amino acid change in the COL6A1, COL6A2 and COL6A3 genes were selected. The frequency of occurrence of variants on any given amino acid was plotted against residue number and shown above the domain schematics of collagen VI. Variants in Glycine residues in the N-terminal region of the collagenous GXY region were coloured on the microfibril model.

### Reporting summary

Further information on research design is available in the Nature Portfolio Reporting Summary linked to this article.

## Data availability

The structures and cryoEM data generated in this study have been deposited in the EMDB and PDB databanks for ColVIα1α2α3$^{ClC2}$ and the local refinement of the collagen VI microfibril under accession codes PDB ID 9GTU; EMD-51567 and PDB ID 9HAN; EMD-51984, respectively. The cryoEM maps for the microfibril double and single beads have been deposited in the EMDB under accession codes EMD-52362 and EMD-52366, respectively. SAXS data for ColVIα1α2α3$^{ClC2}$ have been deposited in SASDBD under accession code SASDWL2. AlphaFold models are provided via figshare at [https://doi.org/10.6084/m9.figshare.28256147]. Source data are provided with this paper. Unless otherwise stated, all data supporting the results of this study can be found in the article, supplementary, and source data files. Source data are provided with this paper.

## Code availability

A custom Python script for visualising the predicted alignment error (PAE) in AlphaFold Multimer predictions is freely available and deposited in the Zenodo database at [https://doi.org/10.5281/zenodo.15880546].

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

## Acknowledgements

We would like to thank Dr Richard Collins in the EM Facility (RRID:SCR_021147; Faculty of Biology, Medicine and Health, University of Manchester) for assistance, and funding from the BBSRC to C.B. (ref: BB/T017643/1) for the Glacios cryoEM used for screening. We thank the Biomolecular Analysis Core Facility (Faculty of Biology, Medicine and Health, University of Manchester) and Dr Tom Jowitt for his assistance. We thank Dr Jordi Bella, University of Manchester, for helpful comments on the manuscript. A.R.F.G., R.D., M.S. are supported by BBSRC funding to C.B. (Ref: BB/V015826/1; BB/V008099/1; BB/Y011740/1) and M.H.B. by a Wellcome PhD studentship (ref: 222780/Z/21/Z). A.M.R. acknowledges funding from Wellcome (ref: 208398/Z/17/Z). We thank the UK National Electron Bio-Imaging Centre (eBIC) for access to cryoEM facilities (ref: BI22724), beamline B21 at Diamond Light Source (ref: MX31850) for BioSAXS data collection and Dr Nathan Cowieson for assistance.

## Author contributions

A.R.F.G. performed all experiments and data analysis, unless stated otherwise. R.D. and M.H.B. performed the mini-collagen VI expression, purification and mutagenesis. M.H.B. performed the BioSAXS analysis. M.S. performed the model building and interpretation. A.M.R. and C.B. designed and supervised the research. A.R.F.G. and C.B. interpreted the data and wrote the manuscript, with input from the other authors.

## Competing interests

The authors declare no competing interests.
