## [Transparent Peer Review file · Nature Communications]

Collagen VI microfibril structure reveals mechanism for molecular assembly and clustering of inherited pathogenic mutations

Corresponding Author: Professor Clair Baldock

Version 1:

Reviewer comments:

Reviewer #1

(Remarks to the Author)

Collagen VI is a complex protein with a multistep intracellular assembly process – triple helical monomers (3 chains), dimers (6 chains) then tetramers (12 chains) - and additional extracellular assembly into microfibrils. Each monomer has three genetically distinct chains: in the main form these are $\alpha 1(\text{VI})$, $\alpha 2(\text{VI})$, $\alpha 3(\text{VI})$. The sequences and interactions that bring the collagen VI protein chains together to form a triple helix are not known, and the regions and amino acids important for dimer, tetramer and microfibril assembly remain unknown or incompletely defined. Dominant and recessive mutations in the three genes, COL6A1, COL6A2, COL6A3 cause the muscle disorders Bethlem myopathy (relatively mild) and Ullrich congenital muscular dystrophy (severe). In addition to pathogenic mutations, collagen VI is very polymorphic, it has many single amino acid variants with unknown significance. This makes clinical diagnosis extremely challenging, particularly when disease causing mutations can be dominant or recessive. Understanding the structure and the interactions important for generating the structure promises to make important contributions to basic cell biology, clinical diagnosis and practice, patient management, and therapeutic approaches. The data presented in this paper, recombinant expression, cryoEM, MALS, SAXS, alpha fold protein structure prediction are extremely high quality and I have no concerns about the data robustness or reproducibility. The figures are clear and the text logically presented. This paper presents important new information about the interactions involved in collagen VI assembly. It is a major advance in our understanding of collagen VI.

My most significant questions/comments are around discussions of pathogenic variants and the criteria the authors have used to determine pathogenicity. Throughout, the authors need to be careful with mutation vs variant terminology and weigh up the evidence for pathogenicity carefully. In cases where the evidence for pathogenicity is not strong (variants of unknown significance for example), the substitutions should be referred to as variants, not mutations.

Introduction - "Polymorphisms in collagen VI genes have been correlated with increased risk of cardiovascular disease, where COL6A3 is a putative causal gene for thoracic aortic aneurysms and dissections (10) and variants in COL6A1 cause Spontaneous Coronary Artery Dissection (11)." The authors should moderate this claim/interpretation. The studies in references 10 and 11 are risk analyses, ref 10 is a GWAS study and ref 11 whole exome sequencing. Leaving a critique of those papers aside, they do not show that collagen VI variants CAUSE the cardiovascular diseases. In fact, in ref 11 only two out of 130 individuals carried a COL6A1 variant (of unknown significance).

Introduction – "Collagen VI monomers interact in antiparallel register..." The monomers are antiparallel but is register the right word? The N-terminal ends of the monomers extend out beyond the overlapping central region in the dimers.

Results – "However, collagen VI is unusual in that it does not have a trimeric C-terminal region..." It's not clear what the authors mean here. Does this mean the C-terminal regions of the three chains don't interact? or don't form disulfide bonds? or are not involved in chain selection?

Results – "suggesting a disulphide bond forms with Cys600($\alpha 2$) in the coiled coil (Figures 6F and 7C). Cys1916($\alpha 3$) is at the bottom of the bead in the region where two collagenous regions within the tetramer contact each other, and an interchain disulphide bond could form between one $\alpha 3$ Cys1916 and the other $\alpha 3$ Cys1916, covalently linking and stabilising the two dimers in the tetramer." It would be helpful here to include the domain locations of the various Cys residues ie which domain Cys600($\alpha 2$) is in, similarly the $\alpha 3$ Cys1916.

It would be helpful to include the domain locations of the various Cys residues throughout, including in the discussion.

Results – "Interestingly, there are two pathogenic mutations that occur in these valine residues (V594I and V598M)(36, 37), so it appears that substitutions to larger hydrophobic residues in the $\alpha 2$ -chain would perturb the internal packing within the trimeric coiled coil, offering a molecular explanation for the disease phenotype." The evidence that these two $\alpha 2$ variants are

pathogenic is scant and the two cited papers don't do any functional analyses. They are both large cohort studies designed to be the first tier of patient diagnosis. There are 87 V594I alleles in gnomAD and it is listed there as "unknown significance". In ClinVar there are three cases, all are called either benign or a VUS, and only one case has an associated phenotype listed (Bethlem myopathy). There are 66 V598M alleles in gnomAD and this variant is not in ClinVar. GnomAD also has 2 V598L alleles and 9 V598A alleles (1 in ClinVar of unknown significance). The data presented here could be consistent with these variants being deleterious but should be discussed in light of the current uncertain nature of their pathogenicity. Discussion – "Our data show that the vWA domains do not appear to be essential for heterotrimerisation and that collagen VI has a short stretch of trimeric coiled coil." The shortest constructs tested included the N1 domains, so how can the authors conclude that the vWA domains "do not appear to be essential"? Perhaps further discussion about how the C1 domains interact (or not) with each other would clarify this.

Figure 8 – The authors don't describe where the list of 'pathogenic missense mutations' comes from, either in the text, the figure legend or the methods. Also, what criteria were used to determine pathogenicity? This is important because I suspect that many of the dots in panel A correspond to variants that are not pathogenic or variants that are classified as VUSs (variants of unknown significance). If just the variants that have been confirmed pathogenic based on a large family pedigree, muscle histology, muscle collagen VI immunostaining, fibroblast biosynthetic data etc were included then regions important for assembly could be more obvious, particularly for the N-terminal end of the triple helix. The N- and C-terminal domains are highly variable with lots of amino acid substitutions of unknown significance so the ones that have strong evidence for pathogenicity could again add to or confirm the structural and interaction data presented here. Including all reported variants likely masks some important relationships.

Figure legend fig 8 - "The y-axis corresponds to the frequency of the occurrence of a mutation on any amino acid...". It's not clear what this means when looking at the figure because there is no y axis scale.

When discussing the implication of the structural models for collagen VI assembly and pathogenic mutations the authors should refer to previous studies that show the effects on assembly. For example, PMID: 18825676 shows that mutations at the N-terminal end of the triple helix affect microfibril formation but not earlier assembly steps. There are a number of mutations in the $\alpha 2$ N2 C-terminal domain that affect assembly. Are these in interacting regions? How does the structural and assembly data fit with PMID: 16613849 which shows microfibril assembly even if the $\alpha 3$ chain only has the C1 domain? What about $\alpha 1$ exon 14 skip mutations that delete Cys345 and prevent dimer formation? The authors should also investigate if variants in the C-terminal interacting faces have been reported in the various databases and/or in patients. A previous study reported the structure of the $\alpha 3$ N2 domain which harbors some pathogenic mutations, see PMID: 32719005. How does this fit with the structures reported here and does the new information help explain the pathogenicity? PMID: 32719005 also showed the structure of an array of $\alpha 3$ N-terminal domains. How does that fit with the current data?

Reviewer #2

(Remarks to the Author)

The manuscript titled "Collagen VI microfibril structure reveals mechanism for molecular assembly and identifies a site for the clustering of inherited pathogenic mutations" by Godwin et al describe the structural characterisation of collagen VI microfibrils by single particle Cryo-EM. Most of the work is sound and the results are impactful for the community. However the article presents major technical problems and should be better presented. The actual version requires a substantial and comprehensive revision. Only after that, it will be possible to review also the details of this work.

Here below comments and concerns:

1) The reconstruction of the Collagen VI microfibril double-bead is technically unsound.

The presented 2D averages (Supplementary Figure 6b) shows a high degree of flexibility between heads. The procedure the authors applied to obtain the 3D reconstruction considered D2 symmetry. However, particles were clearly not symmetric. The FSC plot (Supplementary Figure 6d) the authors reported is unacceptable. The curve does not reach zero. This is concerning and can be diagnostic of both particle duplication or tight masking. Anyway, the resolution estimates are clearly over-optimistic due to overfitting. The local resolution figure they present in Supplementary Figure 6c is meaningless. The highest resolution (marked in blue) is in a featureless blob. This local resolution estimates are unreliable. However, the main take home message of the manuscript does not require this reconstruction to support their claims. So, this reviewer recommends removing this reconstruction.

2) Generally, the methods are very unclear. Methods should be written in a way anybody can reproduce your results.

2.1. under the "CryoEM Data collection and processing" the authors present colVI $\alpha 1\alpha 2\alpha 3$ C1C2 data collection and processing, and also data collection of bovine microfibril, but not the processing, that is in a separated section "Double Bead data set". What is the rational for such a division?

2.2 The description of the data processing is poor. There is a complete lack of important details in the processing.

How many particles they selected after blob picking? What are good particles? How do they extract? What was the pixel size and box size in each extraction? These are key parameters and the report is the standard nowadays in cryo-EM single particle data processing. Please, report.

2.3. The supplementary figure dedicated to the cryo-EM data processing workflow are unrelated to what is written in the methods.

They report a representative micrograph, 2D averages, and sometimes 3D classes. The information is connected through arrows as in a data processing workflow. However, the work flow presented in Supplementary Figure 3, Supplementary Figure 6, Supplementary Figure 7 and Supplementary Figure 8 is far from been representative. In works like this, in which the data processing is not standard is extremely recommended to report a schematic representation of the processing workflow. This can be very helpful to the readers to understand the strategy you choose.

2.4 Truncated FSC plots.

The pixel size for the $\alpha 1\alpha 2\alpha 3$ C1C2 heterotrimer data set is 0.651 Å, while for microfibrils 0.829 Å, that correspond to Nyquist

values of 1.3 Å and 1.6 Å. However, author represented their FSC limit at much lower spatial frequencies. Since there is a complete lack of details in the methods, this reviewer cannot evaluate if it was due to binning at a certain step or if it was deliberately truncated. This is concerning, since FSC plot a high frequencies is important to evaluate overfitting. In fact, this cannot be evaluated in the first reconstruction (mini collagen VI $\alpha 1\alpha 2\alpha 3C1C2$), since their resolution limit is close to the reported Ny.

3) The atomic model of the C-terminal region in the microfibril bead they present is based in a rather low-resolution reconstruction. Even though, two key aspects support their model as mentioned in pg11 ln290; "Thus, aided by the mini-collagen VI structure and the AlphaFold2 prediction...". However, the Authors do not provide any evidence on the correlation of the mini-collagen VI structure or the AlphaFold2 prediction. This is instrumental, since all the claims afterwards are based on the assumption these three sets of data correlate well. Therefore, a figure presenting the overlap, or data on the correlation of these models need to be reported to convince the reader.

Reviewer #3

(Remarks to the Author)

Reviewer #4

(Remarks to the Author)

Godwin et al. reported the cryo-EM structures of human collagen VI $\alpha 1\alpha 2\alpha 3C1C2$ heterotrimer and bovine collagen VI tissue microfibril. They found that the coil-coil region between $\alpha 1$, $\alpha 2$, and $\alpha 3$ is crucial for heterotrimerization. They evaluated mutations in the coil-coil interface residues and discovered that they reduce heterotrimer assembly. They also reconstruct and show the domain organization of the collagen VI microfibril to gain insight into the microfibril assembly.

Specific comments:

1. Authors mentioned chain and domain assignment was explained in further detail in the supplementary methods. However, it was not included.
2. Given the importance of the coil-coil region (Fig. 4B), authors should include the electron density map with the model overlay to see how well the model agreed with the data.
3. It would be helpful to label figure 4 with C1 or C2 to orient readers on which domain interacts with which.
4. Are the mutations of coil-coil region residues sufficient to disrupt the microfibril assembly as well?
5. There are about 300–500 amino acid linkers between the N1 and C1 domains. How certain are they of their assignments that the N1 domain is located next to the C1 domain in the microfibril structure given the low-resolution map (Fig. 7)?
6. It is important to map the pathogenic mutation hotspots in microfibril structure to understand the pathologic etiology; the certainty of the structure assignment is still in question. How do authors cross-validate that their domain assignment and amino acid location are fairly accurate? Are there antibodies or protein markers that can be used to validate the domain location? And it would be useful to include the reconstructed microfibril model.

Version 2:

Reviewer comments:

Reviewer #1

(Remarks to the Author)

All of the concerns I raised in my original review have been appropriately addressed in the revised manuscript and explained in the rebuttal. The additional information in the methods section and supplementary data clarifies the analytical approaches.

Reviewer #2

(Remarks to the Author)

The revised manuscript titled "Collagen VI microfibril structure reveals mechanism for molecular assembly and identifies a site for the clustering of inherited pathogenic mutations" by Godwin et al is significantly improved compared to the previous version. Authors have addressed all our concerns and we deem this manuscript ready for publication.

Reviewer #3

(Remarks to the Author)

Reviewer #4

(Remarks to the Author)

The authors have addressed my concerns in their rebuttal letter. They included model building and domain assignment in the methods section, along with new figures to support their models. Additionally, I reviewed the provided cryoEM map and PDB models and found no issues. Overall, I believe their revisions strengthen the manuscript significantly. The clarity and depth of the new figures, methods, and revised main text enhance the understanding of their findings, making the work more compelling.

Response to reviewers' comments

We thank the reviewers for their comprehensive comments and helpful suggestions. We have addressed each point as follows.

Reviewer #1:

Introduction - "Polymorphisms in collagen VI genes have been correlated with increased risk of cardiovascular disease, where COL6A3 is a putative causal gene for thoracic aortic aneurysms and dissections (10) and variants in COL6A1 cause Spontaneous Coronary Artery Dissection (11)." The authors should moderate this claim/interpretation. The studies in references 10 and 11 are risk analyses, ref 10 is a GWAS study and ref 11 whole exome sequencing. Leaving a critique of those papers aside, they do not show that collagen VI variants CAUSE the cardiovascular diseases. In fact, in ref 11 only two out of 130 individuals carried a COL6A1 variant (of unknown significance).

Thank you for this clarification, this sentence has been reworded as follows: A GWAS study has identified a COL6A3 variant as one risk loci for thoracic aortic aneurysms and dissections (9) and two variants in COL6A1 have been identified in some individuals with Spontaneous Coronary Artery Dissection, although their pathogenic significance is not clear (10).

Introduction – "Collagen VI monomers interact in antiparallel register..." The monomers are antiparallel but is register the right word? The N-terminal ends of the monomers extend out beyond the overlapping central region in the dimers.

We agree the term register could imply the monomers are in alignment rather than overlapping, so have changed this to "Collagen VI monomers interact in a staggered antiparallel arrangement"

Results – "However, collagen VI is unusual in that it does not have a trimeric C-terminal region..." It's not clear what the authors mean here. Does this mean the C-terminal regions of the three chains don't interact? or don't form disulfide bonds? or are not involved in chain selection?

What we meant here is that unlike fibrillar collagens, collagen VI does not have the trimeric C-terminal ColFi domain, which can trimerise independently of the stalk or GXY regions. We have rephrased this statement to "However, collagen VI does not have the trimeric C-terminal ColFi (NC1) domain found in fibrillar collagens"

Results – "suggesting a disulphide bond forms with Cys600(α 2) in the coiled coil (Figures 6F and 7C). Cys1916(α 3) is at the bottom of the bead in the region where two collagenous regions within the tetramer contact each other, and an interchain disulphide bond could form between one α 3 Cys1916 and the other α 3 Cys1916, covalently linking and stabilising the two dimers in the tetramer." It would be helpful here to include the domain locations of the various Cys residues ie which domain Cys600(α 2) is in, similarly the α 3 Cys1916.

We have rephrased this section, which we hope is clearer: "Cys345(α 1) and Cys352(α 2), both in the collagenous region, sit adjacent to the coiled coil region, in an area of density where the coiled coil region contacts the collagenous region. This suggests that a disulphide bond forms between either Cys345(α 1) or Cys352(α 2), in the collagenous region, with Cys600(α 2) from the coiled coil region (Figures 6F and 7C). Cys1916(α 3), also in the collagenous region, is positioned at the bottom of the bead in the region where two collagenous regions within the tetramer contact each other. Thus, an interchain disulphide bond could form between two parallel Cys1916(α 3) residues, covalently linking and stabilising the two dimers in the tetramer (Figure 7C)."

It would be helpful to include the domain locations of the various Cys residues throughout, including in the discussion.

Throughout the results and discussion, the Cys residues described are either in the collagenous region or the coiled coil region and we have now made it clearer in the results and discussion sections, which of these regions the Cys residues are found in.

Results – "Interestingly, there are two pathogenic mutations that occur in these valine residues (V594I and V598M)(36, 37), so it appears that substitutions to larger hydrophobic residues in the α 2-chain would perturb the internal packing

within the trimeric coiled coil, offering a molecular explanation for the disease phenotype.” The evidence that these two $\alpha 2$ variants are pathogenic is scant and the two cited papers don’t do any functional analyses. They are both large cohort studies designed to be the first tier of patient diagnosis. There are 87 V594I alleles in gnomAD and it is listed there as “unknown significance”. In ClinVar there are three cases, all are called either benign or a VUS, and only one case has an associated phenotype listed (Bethlem myopathy). There are 66 V598M alleles in gnomAD and this variant is not in ClinVar. GnomAD also has 2 V598L alleles and 9 V598A alleles (1 in ClinVar of unknown significance). The data presented here could be consistent with these variants being deleterious but should be discussed in light of the current uncertain nature of their pathogenicity.

This sentence has been modified as follows: “Interestingly, there are two variants that can occur in these residues (V594I and V598M), that may be pathogenic (39, 40), could be consistent with larger hydrophobic residues in the $\alpha 2$ -chain perturbing the internal packing within the trimeric coiled coil.”

Discussion – “Our data show that the vWA domains do not appear to be essential for heterotrimerisation and that collagen VI has a short stretch of trimeric coiled coil.” The shortest constructs tested included the N1 domains, so how can the authors conclude that the vWA domains “do not appear to be essential”? Perhaps further discussion about how the C1 domains interact (or not) with each other would clarify this.

This is a good point, our smallest construct still contained the C1 domains so we can’t rule out that they do not have some contribution to trimerisation. However, the C1C2 mini-collagen structure shows no evidence of a trimeric interaction between the C1 or C2 domains. Moreover, there is a high degree of flexibility of the $\alpha 1$ and $\alpha 2$ chain C1 domains so that they are not visible in the EM map, if present, a trimer of these domains should have been visualised. Moreover, we were unable to solve the structure of the C1 mini-collagen, again due to flexibility of these domains, further supporting the lack of a trimeric unit formed by these domains. This is in stark contrast to the NC1 ColFi domain in fibrillar collagens, which can trimerise independently. We’ve rephrased this sentence and added further clarification as follows: “There is a lack of evidence in our structures that the vWA domains form trimeric units. Instead, the C1 domains appear to be mobile in the collagen VI heterotrimer, such that the $\alpha 1$ and $\alpha 2$ chain C1 domains are not visible in the EM map. Rather, the trimeric coiled coil region appears to be important for heterotrimerisation.”

Figure 8 – The authors don’t describe where the list of ‘pathogenic missense mutations’ comes from, either in the text, the figure legend or the methods. Also, what criteria were used to determine pathogenicity? This is important because I suspect that many of the dots in panel A correspond to variants that are not pathogenic or variants that are classified as VUSs (variants of unknown significance). If just the variants that have been confirmed pathogenic based on a large family pedigree, muscle histology, muscle collagen VI immunostaining, fibroblast biosynthetic data etc were included then regions important for assembly could be more obvious, particularly for the N-terminal end of the triple helix. The N- and C-terminal domains are highly variable with lots of amino acid substitutions of unknown significance so the ones that have strong evidence for pathogenicity could again add to or confirm the structural and interaction data presented here. Including all reported variants likely masks some important relationships. Figure legend fig 8 - “The y-axis corresponds to the frequency of the occurrence of a mutation on any amino acid...”. It’s not clear what this means when looking at the figure because there is no y axis scale.

We apologise for the omission of this information, the mutations shown in figure 8 are from the HGMD database and this information has now been added to the methods and the figure legend. The pathogenicity was taken as disease-causing mutations (DM) or probable/possible (DM?) from this database. We have now amended the figure to just include the disease-causing mutations (DM) linked to collagen VI muscular dystrophies and added the missing axes. The legend has been reworded as follows “Frequency plots of missense variants from the HGMD database reported to be disease-causing in collagen VI $\alpha 1$, $\alpha 2$ and $\alpha 3$ chains. The frequency of occurrence of variants on any given amino acid is shown above the domain schematics of collagen VI, approximately to scale, showing in which domains the variants occur. The number of triangles on the y-axis corresponds to the frequency of the occurrence of a mutation on any amino acid.”

When discussing the implication of the structural models for collagen VI assembly and pathogenic mutations the authors should refer to previous studies that show the effects on assembly.

Thank you for highlighting these papers, we have added discussion of these papers to the manuscript as detailed below each point.

For example, PMID: 18825676 shows that mutations at the N-terminal end of the triple helix affect microfibril formation but not earlier assembly steps.

Thank you for highlighting this publication and the effect of mutations in the N-terminal collagenous region on collagen VI assembly. We have incorporated this into the discussion: "A previous study has shown that mutations in glycine residues at the N-terminal end of the triple helix compromised intracellular assembly and disulphide bonding of tetramers. Dimers containing mutant chains were able to assemble into tetramers, but stabilisation of the tetramers by disulphide bonding was compromised and microfibril formation was impaired (36). This suggests that these mutations can impair head-to-tail dimerisation and formation of the disulphide bond that stabilises the dimer, and also formation of higher order assemblies."

There are a number of mutations in the $\alpha 2$ N2 C-terminal domain that affect assembly. Are these in interacting regions?

We think this point relates to the $\alpha 2$ C2 domain (as the $\alpha 2$ -chain does not contain an N2 domain), and the C2 domain contains a number of mutations. We have addressed this point below in the query about variants in the C-terminal interacting faces.

How does the structural and assembly data fit with PMID: 16613849 which shows microfibril assembly even if the $\alpha 3$ chain only has the C1 domain?

As our microfibrils are mature, they are lacking the $\alpha 3$ C2-C5 domains, removed by furin/BMP1 processing, so we cannot comment on their role in the assembled microfibril. PMID: 16613849 (Lamande et al., 2006) indicates that although monomers, dimers and tetramers can form when the $\alpha 3$ chain only has the C1 domain, assemblies of tetramers were infrequently observed for $\alpha 3(\text{VI})$ N6-C1 and N6-C2 suggesting that the C-terminal domains support microfibril formation. Our data are consistent with this observation, as the mini-collagen can trimerise in the absence of the C2 domains but in the mini-collagen C1C2 structure, the $\alpha 3$ C1 and C2 domains are involved in interactions with the $\alpha 2$ -chain which may be stabilising interfaces required for higher order assembly.

What about $\alpha 1$ exon 14 skip mutations that delete Cys345 and prevent dimer formation?

Thank you for highlighting these mutations. The lack of dimer formation with deletion of $\alpha 1$ Cys345 reported in Lamande' et al. 1999. JBC would be entirely consistent with the location of this residue in the triple helical region adjacent to the cysteine-rich coiled coil region. This provides further support for the requirement of a disulphide bond between the collagenous region and the coiled coil region for dimerisation (Figure 7Cii). We have added the following sentence to the discussion indicating this. "Indeed, Bethlehem myopathy mutations that skip exon 14 in the $\alpha 1$ chain deleting cys345 prevent dimer formation (53), which suggests this cysteine in the $\alpha 1$ chain collagenous region is bonding to the coiled coil region (Figure 7Cii)."

The authors should also investigate if variants in the C-terminal interacting faces have been reported in the various databases and/or in patients.

We checked the location of pathogenic variants in the HGMD in the C-terminal domains in the microfibril structure. Many of the residues were located in secondary structural elements and may effect folding and secretion of the protein, but three variants in the ($\alpha 2$)C2 domain lie in interfaces with the ($\alpha 3$)C1 and ($\alpha 1$)C2 domains. A new panel E has been added to Figure 8 showing the location of these variants and the following text added to the results. "The ($\alpha 2$) chain C-terminal domains contain a number of pathogenic variants, three of which mapped onto interfaces between the ($\alpha 2$)C2 domain and the ($\alpha 3$)C1 domain within the heterotrimer and the ($\alpha 1$)C2 domain within the tetramer interface (Figure 8E)."

A previous study reported the structure of the $\alpha 3$ N2 domain which harbors some pathogenic mutations, see PMID: 32719005. How does this fit with the structures reported here and does the new information help explain the pathogenicity? PMID: 32719005 also showed the structure of an array of $\alpha 3$ N-terminal domains. How does that fit with the current data?

Unfortunately, due to the flexibility of the $\alpha 3$ N2-N9 domains, they are not visible in our microfibril structure. Solomon-Defega et al 2020 (PMID: 32719005) and prior to that Beecher et al., 2011 and Maas et al., 2016 show that the array of

N-terminal domains in the $\alpha 3$ chain are flexible which is consistent with their lack of resolution in our structure. We have added these citations to the results where we describe the density for the N2 domain and flexibility of the N2-N9 region. There is low resolution density for the $\alpha 3$ N2 domain protruding from the side of the bead (Figure 7B(ii)), but this domain is not interacting with the other domains visible in the microfibril reconstruction. Although a minimal six N-terminal domains are required for microfibril assembly, we do not observe these domains in the microfibril reconstruction, which would suggest that even though they are essential for microfibril formation, they are not stabilised within the microfibril structure once the microfibril has formed. The pathogenic mutations reported in $\alpha 3$ N2 domain in Solomon-Defega et al 2020 and Sasaki et al., 2000 appeared to prevent folding and secretion of the protein or resulted in abnormal dimerisation in vitro.

Reviewer #2

1) The reconstruction of the Collagen VI microfibril double-bead is technically unsound. The presented 2D averages (Supplementary Figure 6b) shows a high degree of flexibility between heads. The procedure the authors applied to obtain the 3D reconstruction considered D2 symmetry. However, particles were clearly not symmetric. The FSC plot (Supplementary Figure 6d) the authors reported is unacceptable. The curve does not reach zero. This is concerning and can be diagnostic of both particle duplication or tight masking. Anyway, the resolution estimates are clearly over-optimistic due to overfitting. The local resolution figure they present in Supplementary Figure 6c is meaningless. The highest resolution (marked in blue) is in a featureless blob. This local resolution estimates are unreliable. However, the main take home message of the manuscript does not require this reconstruction to support their claims. So, this reviewer recommends removing this reconstruction.

We would like to thank the reviewer for highlighting these points and for opportunity to provide further detail and clarification on the cryoEM data processing.

The reviewer is correct that the reconstruction of the double bead is not key to the message of the manuscript, and was included to illustrate where the sub-regions of the microfibril originate from and to generate the illustrative bead model shown in Figure 7C(i). We hope we have satisfactorily addressed the issues highlighted above, by reprocessing the bovine double bead reconstruction using the straightest most homogenous particles, as the reviewer correctly highlighted that flexibility along the microfibril axis breaks the D2 symmetry. The processing is detailed in the revised workflow (Supplementary Figure 8). The symmetry was relaxed in the latter stages as detailed in the methods. We also removed duplicate particles and the FSC curve now reaches zero with resolution estimate of 12.3 Å, and the local resolution figure shows resolution estimates around this range (Supplementary Figure 11). The updated double bead map is shown in Figure 7C(i), however the reprocessing of the data did not change any findings or conclusions in the manuscript.

2) Generally, the methods are very unclear. Methods should be written in a way anybody can reproduce your results. We have rewritten methods section for the cryoEM data processing to address the reviewer's points and have now made the methods section more comprehensive in its description of the cryoEM data processing as detailed in the response to each point below.

2.1. under the "CryoEM Data collection and processing" the authors present colVI $\alpha 1\alpha 2\alpha 3$ C1C2 data collection and processing, and also data collection of bovine microfibril, but not the processing, that is in a separated section "Double Bead data set". What is the rational for such a division?

We have improved the organisation and flow of the methods section with separate sections for data collection and processing of the mini-collagen and for the data collection and processing of the microfibril data.

2.2 The description of the data processing is poor. There is a complete lack of important details in the processing. How many particles they selected after blob picking? What are good particles? How do they extract? What was the pixel size and box size in each extraction? These are key parameters and the report is the standard nowadays in cryo-EM single particle data processing. Please, report.

We have provided more detail of the data processing to include the requested details for each step, this includes the number of particles picked at each stage of particle picking, how many particles were selected from each processing

step and the criteria used to decide whether a particle was good in 2D classification. We have reported the extraction box size and pixel size of the data set after each extraction job.

2.3. The supplementary figure dedicated to the cryo-EM data processing workflow are unrelated to what is written in the methods. They report a representative micrograph, 2D averages, and sometimes 3D classes. The information is connected through arrows as in a data processing workflow. However, the work flow presented in Supplementary Figure 3, Supplementary Figure 6, Supplementary Figure 7 and Supplementary Figure 8 is far from been representative. In works like this, in which the data processing is not standard is is extremely recommended to report a schematic representation of the processing workflow. This can be very helpful to the readers to understand the strategy you choose.

We have provided much more detailed schematic representations of the data processing workflows for each dataset, shown in Supplementary Figures 3 and 8, 9 and 10 to give the reader a better understanding of how the data was processed.

2.4 Truncated FSC plots. The pixel size for the $\alpha 1\alpha 2\alpha 3C1C2$ heterotrimer data set is 0.651 Å, while for microfibrils 0.829 Å, that correspond to Nyquist values of 1.3 Å and 1.6 Å. However, author represented their FSC limit at much lower spatial frequencies. Since there is a complete lack of details in the methods, this reviewer cannot evaluate if it was due to binning at a certain step or if it was deliberately truncated. This is concerning, since FSC plot a high frequencies is important to evaluate overfitting. In fact, this cannot be evaluated in the first reconstruction (mini collagen VI $\alpha 1\alpha 2\alpha 3C1C2$), since their resolution limit is close to the reported Ny.

The data was binned which is why the FSC plots could appear truncated. We have rewritten the methods to add the box size and down-sampling used throughout the cryoEM processing and for each of the final maps. These details have also been added to the figure legends of Supplementary Figures 4, 11, 12 and 13. As the resolution limit is close to the Nyquist frequency in the mini-collagen VI map, in Supplementary Figure 4B(ii) we have included the FSC plot from a reconstruction of unbinned particles, with the alignments of the final binned reconstruction, in order to address the reviewer's concerns.

3) The atomic model of the C-terminal region in the microfibril bead they present is based in a rather low-resolution reconstruction. Even though, two key aspects support their model as mentioned in pg11 ln290; "Thus, aided by the mini-collagen VI structure and the AlphaFold2 prediction...". However, the Authors do not provide any evidence on the correlation of the mini-collagen VI structure or the AlphaFold2 prediction. This is instrumental, since all the claims afterwards are based on the assumption these three sets of data correlate well. Therefore, a figure presenting the overlap, or data on the correlation of these models need to be reported to convince the reader.

As suggested, we have added a new Supplementary Figure 14, which shows the atomic models for the mini-collagen VI $\alpha 1\alpha 2\alpha 3C1C2$ and bovine microfibril C-terminus, and Alphafold2 model of $\alpha 1\alpha 2\alpha 3C1C2$ overlaid. We also performed Alphafold3 predictions, which was available after the structures were determined, and which correlates even better than the Alphafold2 prediction. In this figure, we have also included pairwise RMSD values comparing all the models to each other showing the correlation of the models. We hope this figure shows convincingly that the mini-collagen structure correlates well with the equivalent domains in the microfibril, supporting the model building of the microfibril structure. We have added a sentence to the results section "Supplementary Figure 14 shows the overlay of the coiled coil region, $\alpha 3$ C1 and $\alpha 2$ C2 domains between the mini-collagen and bovine microfibril structures."

Reviewer #4:

1. Authors mentioned chain and domain assignment was explained in further detail in the supplementary methods. However, it was not included.

We apologise this document was not accessible for review. We have now expanded the methods, to address the comments of reviewer 2, and have provided more detail on the model building and incorporated this information into the main methods section rather than supplementary information.

2. Given the importance of the coil-coil region (Fig. 4B), authors should include the electron density map with the model overlay to see how well the model agreed with the data.

Figure 4A showed the cryoEM density and model overlay for the coiled coil region separated into the three individual α -chains for clarity. We have now also added to Supplementary Figure 5 further images for the density of the coiled coil region with the models overlaid.

3. It would be helpful to label figure 4 with C1 or C2 to orient readers on which domain interacts with which.

Labels for α 3C1 and α 2C2 have now been added to Figure 4 panels D and E.

4. Are the mutations of coil-coil region residues sufficient to disrupt the microfibril assembly as well?

Unfortunately, we are not able to test this experimentally, as it is not currently possible to make microfibrils recombinantly in order to introduce mutations. The microfibrils we have imaged are extracted from mammalian tissue, which does not make it possible to readily test the impact of mutations on microfibril assembly. Currently, this is only possible if samples are available from patients with disease-causing mutations. This limitation was our rationale for generating the mini-collagen expression system to provide a means to introduce and test the impact of mutations on the formation of the heterotrimer. We would predict that mutations in the coiled coil that perturb the formation of heterotrimers would also disrupt microfibril formation.

5. There are about 300–500 amino acid linkers between the N1 and C1 domains. How certain are they of their assignments that the N1 domain is located next to the C1 domain in the microfibril structure given the low-resolution map (Fig. 7)?

6. It is important to map the pathogenic mutation hotspots in microfibril structure to understand the pathologic etiology; the certainty of the structure assignment is still in question. How do authors cross-validate that their domain assignment and amino acid location are fairly accurate? Are there antibodies or protein markers that can be used to validate the domain location?

The reviewer's points 5 and 6 relate to the certainty of domain locations in the single bead map and microfibril bead model, shown in Figures 7 and 8B respectively. To address these points, we have now provided more explanation of the model building and domain assignment in the methods, and we have included additional Supplementary Figures (14 and 15) showing the overlay of the mini-collagen and microfibril maps and structures to cross-validate the structure of the microfibril C-terminal region. We have summarised these details below and describe published data that underpin these domain assignments.

The mini-collagen system was designed to aid interpretation of the microfibril structure and the resolution of this 3.14 Å map allowed unambiguous identification of the domains in this map from sidechain densities. Map-map correlation with the head part of the 4.33 Å bovine microfibril map showed the absence of the α 3C2 domain (Supplementary Figure 15A), which is cleaved upon microfibril maturation (Heumuller et al., 2019; Aigner et al., 2002). The α 3C1 and α 2C2 domains and coiled coil region, present in both the mini-collagen and the head part of the microfibril were in the same relative locations in the two maps (Supplementary Figures 14 and 15A). In each asymmetric unit, there is then an additional two C1 domains and three N1 domains, which were docked into the single bead map. For the C1 domains, in the 4.33 Å map, we see clear connectivity linking the C1 and C2 domains for both the α 1 and α 2 chains in the head part of the structure making this assignment clear, and also consistent with their relative locations in AlphaFold2 and AlphaFold3 predictions (Supplementary Figure 14 and 15B).

As the reviewer highlights, there are 336 amino acids in the collagenous region of each chain, spanning the N-terminal to C-terminal region, which have a uniform triple-helical structure. However, the microfibril single bead map in Figure 7 (and full bead in Figure 8B) do not show the full microfibril period, rather the N1 domains located adjacent to the C1 domains are contributed from an adjacent tetramer. For clarity, we have added a schematic to figure 8B to highlight which region of the full microfibril this relates. We do understand the reviewers point though, regarding why the N1 domains were modelled into this region of the single bead map. There are prior biochemical studies and antibody labelling that underpin these domain placements ((Furthmayr et al., 1983; Engel et al., 1985; Perris et al 1993). In brief,

these studies denatured microfibrils to tetramers, followed by reduction to dimers and monomers. Each of these oligomeric forms were treated with proteases and/or collagenases to determine their composition and location of N- and C-terminal regions in microfibrils and oligomers. Previous antibody mapping was performed to locate specific domains within the tetramer, which are consistent with the C-terminal domains forming the head or outer region of the bead and the N-terminal domains the inner bead region (Perris et al., 1993).

We found that the placement of each of the N1 domains from the three alpha-chains within this region of the single bead map was consistent with the relative domain model positioning predicted by AlphaFold. This arrangement also fitted with the start of the density for the triple-helical region, that emerges from the C-terminus of these domains, and the density for the $\alpha 3$ N2 domain. We now show these aspects in new Supplementary Figure 15C. For these reasons, we are confident in the domain and chain assignments in the single bead. However, at $\sim 6\text{\AA}$ resolution we cannot unequivocally rule out that the α -chain assignments for the N1 domains could be assigned differently (e.g. $\alpha 1$ N1 could be $\alpha 2$ N1), but this would not change the positioning of the triple-helical glycine mutations highlighted in Figure 8B. We have expanded both the detail of our description of the fitting of the N-terminal region and have qualified the certainty of the fitting with the following wording. "Based on earlier biochemical studies and antibody-labelling (Furthmayr et al., 1983; Engel et al., 1985; Perris et al., 1993), the remaining vWA domains at the bottom of the bead are likely to be the N1 domains from the three α -chains. An AlphaFold model of the N1 domains from all three chains was docked at the bottom of the bead, which fitted well with the cryoEM density (Supplementary Figure 15). The model fitted in only one orientation with the $\alpha 1$ and $\alpha 2$ N1 domains above and the $\alpha 3$ N1 domain underneath (Figure 7B), which placed the beginning of the triple-helical region in the appropriate density. When docked in this orientation, at higher threshold values there is extra density connected to the $\alpha 3$ N1 domain, which would correspond to the flexible $\alpha 3$ -chain N2-N9 domains. However, in the absence of a high-resolution map for this region, we cannot exclude the possibility that the N1 domains from the three chains could be assigned differently in this region, but this would not substantially change the modelling described below." We hope this now provides a more complete explanation for the locations of specific domains and the certainty of their assignments.

And it would be useful to include the reconstructed microfibril model.

We have uploaded the pdb file for the microfibril model to the figshare folder for the review process. This is an illustrative model building on the microfibril C-terminal structure with rigid body docking of the C1 and N1 domains and stretches of triple-helical region. Due to the lower resolution of the microfibril bead map, the model has not undergone refinement to the cryoEM density so there are some chain breaks and geometrical irregularities in the GXY region. For this reason, we have not deposited the model to a database, but are happy to make it available upon request so we can provide together with appropriate caveats.

References

- Aigner T., Hambach, L., Soder, S., Schlotzer-Schrehardt, U., and Poschl, E. (2002) The C5 domain of Col6A3 is cleaved off from the Col6 fibrils immediately after secretion. *Biochemical and biophysical research communications* 290, 743-748.
- Engel, J., Furthmayr, H., Odermatt, E., von der Mark, H., Aumailley, M., Fleischmajer, R., and Timpl, R. (1985) Structure and macromolecular organization of type VI collagen. *Ann N Y Acad Sci* 460, 25-37.
- Furthmayr H., Wiedemann, H., Timpl, R., Odermatt, E., and Engel, J. (1983) Electron-microscopical approach to a structural model of intima collagen. *Biochem J* 211, 303-311.
- Heumuller S. E., Talantikite, M., Napoli, M., Armengaud, J., Morgelin, M., Hartmann, U., Sengle, G., Paulsson, M., Moali, C., and Wagener, R. (2019) C-terminal proteolysis of the collagen VI alpha3 chain by BMP-1 and proprotein convertase(s) releases endotrophin in fragments of different sizes. *J Biol Chem* 294, 13769-13780.
- Perris R., Kuo, H. J., Glanville, R. W., Leibold, S., and Bronner-Fraser, M. (1993) Neural crest cell interaction with type VI collagen is mediated by multiple cooperative binding sites within triple-helix and globular domains. *1993 Experimental Cell Research*. 1993, 209:103-117.